# Two decades of distributed global radiation time series across a mountainous semiarid area (Sierra Nevada, Spain)

Cristina Aguilar[1], Rafael Pimentel[1], María J. Polo[1]

[1]Fluvial Dynamics and Hydrology Research Group, Andalusian Institute of Earth System Research, University of Cordoba, Cordoba, Spain

*Correspondence to*: Cristina Aguilar (caguilar@uco.es)

**Abstract.** The main drawback of the reconstruction of high resolution distributed global radiation ($R_g$) time series in mountainous semiarid environments is the common lack of station-based solar radiation registers. This work presents nineteen years (2000-2018) of high spatial resolution (30 m) daily, monthly, and annual global radiation maps derived using the GIS-based model proposed by Aguilar et al. (2010) in a mountainous area in southern Europe: Sierra Nevada (SN) Mountain Range (Spain). The model was driven by in situ daily global radiation measurements, from sixteen weather stations with historical records in the area, a 30 m digital elevation model and 240 cloud-free Landsat images. The applicability of the modeling scheme was validated against daily global radiation records at the weather stations. Mean RMSE values of 2.63 MJ m$^{-2}$ day$^{-1}$ and best estimations on clear-sky days were obtained. Daily $R_g$ at weather stations revealed greater variations in the maximum values but no clear trends with altitude in any of the statistics. However, at the monthly and annual scales, there is an increase in the high extreme statistics with the altitude of the weather station, especially above 1500 m a.s.l. Monthly $R_g$ maps showed significant spatial differences of up to 200 MJ m$^{-2}$ month$^{-1}$ that clearly followed the terrain configuration. July and December were clearly the months with the highest and lowest values of $R_g$ received and the highest scatter in the monthly $R_g$ values was found in the spring and fall months. The monthly $R_g$ distribution was highly variable along the study period (2000-2018). Such variability, especially in the wet season (October-May), determined the inter annual differences of up to 800 MJ m$^{-2}$ year$^{-1}$ in the incoming global radiation in SN. The time series of the surface global radiation datasets here provided can be used to analyze inter-annual and seasonal variation characteristics of the global radiation received in SN with high spatial detail (30 m). They can also be used as cross-validation reference data for other global radiation distributed datasets generated in SN with different spatio-temporal interpolation techniques. Daily, monthly, and annual datasets in this study are available at https://doi.pangaea.de/10.1594/PANGAEA.921012 (Aguilar et al., 2021).

## 1 Introduction

High mountain areas in semiarid environments present singular characteristics due to the continuous interaction of alpine conditions in the summits with the surrounding semiarid climate. They play a key role as water providers during the warm and dry season when they often constitute the only water source for many rivers. Here, water fluxes from the snowpacks show a shift from the predominant partition between snowmelt and sublimation usually found in colder and wetter climates on an annual and seasonal basis (Herrero and Polo, 2016). This shift is caused by the radiation balance that enhances sublimation during cold and dry periods and intense snowmelt rates during late winter and spring in these areas (McDonell et al., 2013; Liu et al., 2019). However, weather stations are not always equipped to monitor the global radiation nor their components and, moreover, they are seldom found in high altitudes, especially over 1500 m a.s.l., which makes it difficult to accurately assess not only the solar radiation temporal regime but also the spatial patterns of solar radiation fields in high mountain areas. This impacts the availability of data for studies in mountains dealing with climate and hydrology, global warming, ecosystem services provided by the snow areas, and environmental and social and economic impacts on-site and downstream (Yang et al., 2010; Liu et al., 2012a; Tang et al., 2019). It is not surprising that many mountain regions are identified as biodiversity hotspots around the world, with Mediterranean and other semiarid to arid regions being highly represented (Myers et al., 2000; O'Farrell et al., 2010; Hewitt, 2011; Pauli et al., 2012).

There are several research papers on solar radiation estimations from routine ground-based observations in high altitude regions (Dubayah and van Katwijk, 1992; Dubayah, 1994; Tovar et al., 1995; Oliphant et al., 2003; Tovar-Pescador et al., 2006; Yang et al., 2006, 2010; Batllés et al., 2008; Bosch et al., 2008; Sheng et al., 2009; Aguilar et al., 2010; Mamassis et al., 2012; Chen et al., 2013; Zhang et al., 2020). All of them insist on the need to consider topographic effects and advise of the errors that simple interpolation/extrapolation techniques can create. Radiation data obtained from a dense and properly-maintained weather station network in mountainous areas are rarely available and therefore, modeling techniques need to be applied. Liu et al. (2012a) state that the most difficult issue in solar radiation modeling in data sparse regions is cloud accounting, due to the rapid spatially and temporally changing weather conditions and the three-dimensional structure of clouds. This complexity adds to the heterogeneity resulting from shadowing and reflection due to steep topography (Dubayah, 1992; Batllés et al., 2008; Mamassis et al., 2012; Chen et al., 2013; Zhang et al., 2019, 2020).

According to Dubayah and Rich (1995), as solar radiation models become more complex, they can be more difficult to use, mainly because of the requirement for additional input data. In fact, the complexity of physically-based solar radiation formulations for topography and the lack of the data needed to drive such formulations led in the past to the lack of suitable modeling tools (Dubayah, 1994). Thus, it is important that the models allow for some flexibility regarding the component of radiation calculated and the input data needed.

Excluding traditional interpolation methods there are two major methods for solar radiation modeling, namely, satellite-derived solar radiation estimates, and Geographic Information Systems (GIS)-based solar radiation models. Satellite-derived solar radiation models provide a wide spatial and temporal coverage, but low spatial resolution when dealing with pixels with a

strong topographic gradient. By contrast, GIS-based models calculate the incoming solar radiation for each cell of a digital elevation model (DEM) and allow for higher spatial resolutions including topographic effects. In the past decades, several models based on GIS have been proposed (e.g., Dubayah and Rich, 1995; Fu and Rich, 2000a, 2002; Wilson and Gallant, 2000; Goldberg and Häntzschel, 2002; Sùri and Hofierka, 2004; Liu et al., 2012a; Zhang et al., 2019, 2020). Required input data include digital elevation values and atmospheric attenuation parameters that are commonly estimated from ground-based measurements and/or satellite data (Dubayah, 1994).

The aim of this study was to generate the spatiotemporal distribution of global solar radiation in a high mountain semiarid area in southern Spain with a modeling scheme that reconstructs time map series from the usually available weather datasets. For this purpose, a GIS-based topographic solar radiation model (Aguilar et al., 2010) was applied in Sierra Nevada (SN) (Spain), a high mountain range running west-east parallel to the Mediterranean coastline with influence from both the sea and the African continent to the South, and the continental conditions to the North. The accuracy of solar radiation estimates by the model were evaluated in terms of the error in the approximation to observed data. This study site is a high-value environmental area declared Biosphere Reserve by UNESCO in 1986 due to the exceptional presence of endemisms (Heywood, 1995; Blanca et al., 1998; Anderson et al., 2011; Cañadas et al., 2014). Besides, SN is also included in the Global Change Observatories Network given its singular location between two seas and two continents, and its extreme topographic gradients (Bonet-García et al., 2015).

This paper presents 19 years of daily, monthly, and annual solar radiation maps with high resolution (30 m) over SN. The huge number of members involved in the management of this area make this information valuable in different fields, such as: hydrology, crucial role of energy budget in the hydrological cycle over this area; ecology, ecological communities' behaviour and development clearly link with the amount of energy available; production systems downstream, as hydropower facilities and traditional to tropical crop systems from the top to downhills. Besides, these data sets directly contribute, or are relevant for many studies that could do so, to two of the 23 Unsolved Problems in Hydrology (UPH) recently posed by Blöschl et al. (2019) in a participatory analytical discussion among the scientific community: UPH 16 "How can we use innovative technologies to measure surface and subsurface properties, states and fluxes at a range of spatial and temporal scales?" and UPH 5 "What causes spatial heterogeneity and homogeneity in runoff, evaporation, subsurface water and material fluxes (carbon and other nutrients, sediments), and in their sensitivity to their controls (e.g. snowfall regime, aridity, reaction coefficients)?".

## 2 Study site

The Sierra Nevada mountain range (SN) is located 35 km north from the Mediterranean Sea (Fig. 1) and constitutes a mountainous area of the Natura 2000 network. Elevations rise up from 262 m a.s.l. to 3479 m a.s.l. in a 4583.72 km$^2$ area that runs parallel to the sea. High altitudinal gradients are representative of the area, with variation in elevation of about 3400 m in less than 40 km of horizontal distance and a mountain climate in the summits surrounded by Mediterranean climate in the

lower areas. Thus, the interaction of such conditions creates a strong heterogeneity in terms of soil types, landforms and vegetation species that determine a complex hydrological response in the area and many endemic species (Heywood, 1995; Blanca et al., 1998; Anderson et al., 2011). The rainfall regime is highly variable, even in consecutive years, with annual cumulative values in the period (1960-2000) that range between 200 mm in dry years to 1000 mm in wet years, with an average value of 510 mm (Pérez-Palazón et al., 2015). Temperature regime is also heterogeneous, with values of 26, 12.5 and 0.4 ºC, for maximum, mean, and minimum daily temperature in the same period.

The snow presence becomes relevant from November above 2000 m a.s.l. and extends up to spring with conditions that make it possible the activity of a major ski resort in the area. However, in some winters, mild episodes can be found in January and February that melt most of the snow much earlier than the mean end of the snow season in the area (Herrero et al., 2009; Herrero and Polo, 2012). Because of its singular characteristics and fragile environment, Sierra Nevada receives international recognition as a Biosphere Reserve (1986), a National Park (1999), an Important Bird Area (2003), a Special Area of Conservation (2012) and one of the International Global Change Observatories in Mountain Areas. These environmental protection figures together with the different and numerous members involved in the management of such a unique area have determined the strong effort in data collection in the last years to advance in the knowledge of the different aspects that determine the dynamics of this natural system. Moreover, global warming impacts threaten the environmental values of this system but also the associated ecosystem services and social and economic activities due to the estimated shift of the snowfall regime (Pérez-Palazón et al., 2018).

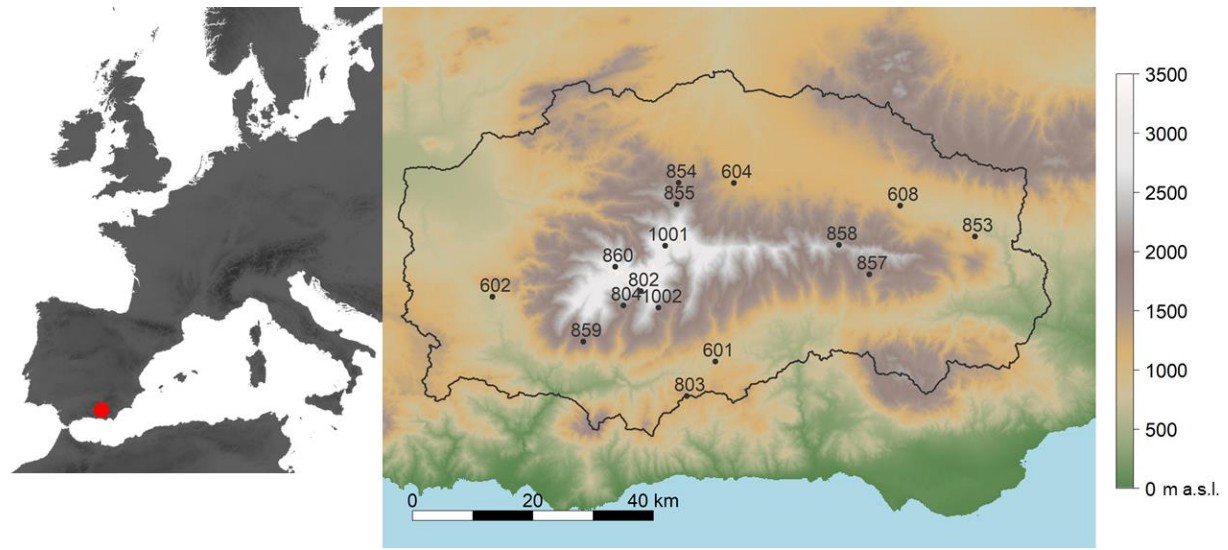

**Figure 1. Location of the study site in southern Spain (left). Digital Elevation Model (DEM) and weather stations in Sierra Nevada (SN) (right). The numbers correspond to the station codes.**

## 3 Data

### 3.1 Input data

A digital elevation model (DEM) with 30 m spatial resolution and 1m vertical precision was used in this study (Fig. 1). The DEM was provided by the Andalusian regional administration and it was generated by digital stereo correlation of aerial photographs of the Spanish National Plan of Orthophotography. The DEM is used to calculate the slope, aspect, sky view factor and terrain configuration maps that are used in the modeling process (Dozier and Frew, 1990).

Meteorological input data are the longest available in-situ daily global radiation ($R_{go}$) of 16 weather stations over the area (Fig. 1 and Table 1). The extent of the records in all weather stations ($N_o$ in Table 1) was considered long enough to carry out the evaluation process dating from February 2000 for the oldest station (608 in Table 1). 12 out of the 16 weather stations are located above 1500 m a.s.l. and 7 of them above 2000 m a.s.l. (Fig. 1). The stations belong to four different organizations: The Department of Agriculture, Fisheries and Environment of the Andalusian Government (601-608 in Table 1), the Water and Environment Agency (1001 and 1002 in Table 1), the National Parks Organization (853-860 in Table 1) and the Guadalfeo Network (802-804 in Table 1) described in Polo et al. (2019). Pyranometers used to collect the data were of different natures but all of them with a characteristic range of around $0.35 \sim 1.1$ μm: Skye SP1110 (stations 601, 602, 604 and 608), Kipp & Zonen SP-Lite pyranometer (station 802), HuksefluxLP02 (station 803), HuksefluxNR01 (stations 1001, 1002 and 804) and Middleton Net Solar CNR1 (stations 853, 854, 855, 857, 858, 859 and 860).

In order to generate the complete global radiation data series for the whole-time span (01/02/2000-31/12/2018) we first apply a quality-control check to the recorded data at the weather stations.

### 3.2 Data quality control

Numerous studies on quality control of measured solar radiation data can be found in the literature (Geiger et al., 2002; Younes et al., 2005; Moradi, 2009; Journée and Bertrand, 2011). Compared to other meteorological variables, solar radiation measurement is more prone to errors (Moradi, 2009). Younes et al. (2005) state two main sources of errors related to in situ measurement of solar radiation: those related to equipment and uncertainty and operational errors. Thus, prior to any computation two basic screenings were applied to recorded daily global radiation data to discard suspicious records associated with equipment and operational errors (Younes et al., 2005).

1. Observed daily global radiation ($R_{go}$) must be between the daily extraterrestrial radiation ($R_{ext}$) and a minimum 3% of $R_{ext}$ (Geiger et al., 2002; Moradi, 2009).

2. Observed daily global radiation ($R_{go}$) must be lower than the clear daily global radiation ($R_{gcs}$) observed under a highly transparent clear sky (Wu et al., 2007). $R_{gcs}$ values were calculated with the model developed by Ineichen and Perez (2002) and the parameterization of Kasten and Young (1989) for the air mass. More detail regarding the equation as well as its parameters can be found in Aguilar et al. (2010).

The excluded values from these tests did not reach 1% of the data at any weather station.

A third quality control screening was applied following Younes et al. (2005) to detect erroneous data due to operational errors
related with particularities of weather stations in high altitudes (e.g., shadows, impacts of snow, mechanical failures, etc.).
They suggest a semi-automatic procedure that allows the creation of an expectancy envelope in the clearness index (CI)-diffuse
to global irradiance ratio (k) domain to reject data too obviously erroneous. The CI data range is divided into bands of equal
width, within which the mean and standard deviation of the k values, $\mu_k$ and $\sigma_k$, are calculated. The top and bottom boundary
shapes are identified by fitting two polynomials through the points $\mu_k \pm b\sigma_k$ limited between 0 and 1 to respect the physical
range of the CI. In this study b values between 2 and 3 were applied in order to limit both, the rejection of good data and the
acceptance of erroneous data to small percentages.
The CI was calculated with the observed data at each weather station. However, no measurements of daily diffuse radiation,
$R_d$, were available. Thus, the model proposed by Aguilar et al. (2010) was applied to generate daily diffuse radiation ($R_{dp}$) at
each weather station without considering the observed global data at such station. Obviously, this assumption depends on the
validity of the model as well as on the quality of $R_{go}$ datasets at the remaining weather stations. However, under the common
lack of diffuse solar radiation measurements like the present one, modeling them can be an alternative (e.g., Yang et al., 2020)
to reject erroneous $R_g$ observations. This approach was proposed once the model had already been validated in a previous study
(Aguilar et al., 2010) but keeping in mind the intrinsic limitations and assumptions previously stated.
After this quality test, the percentage of excluded values did not reach 10% at any weather station, with a mean value close to
2% when the whole set of stations was considered. Table 1 shows selected descriptors of the data sets at each station in this
study after all the quality check process and Figure 2 shows the chronogram of the final input data availability per station (N
in Table 1) used in this study.

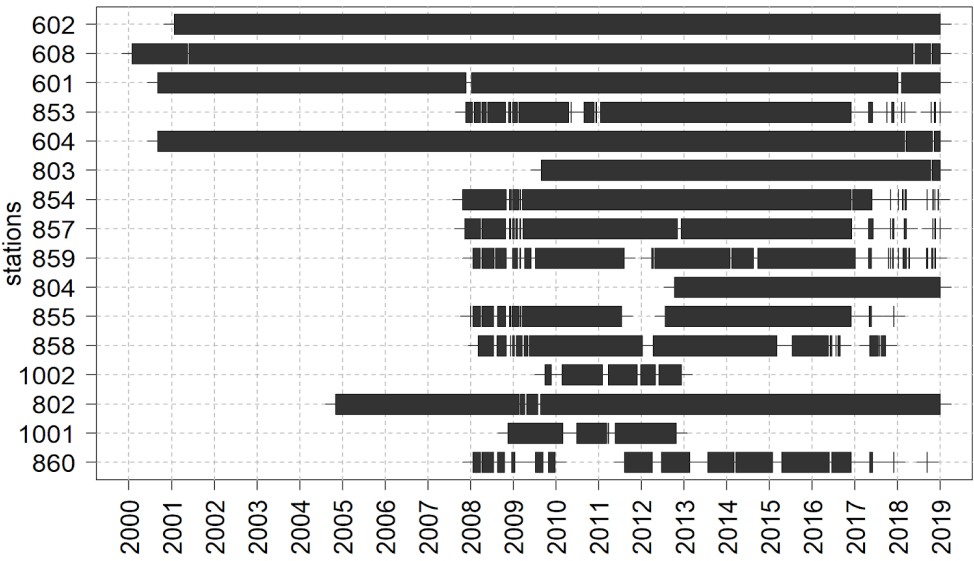


**Figure 2. Data availability in the analyzed period (01 Feb 2000 - 31 Dec 2018) for each weather station. Stations are sorted by**
**increasing altitude from the top to the bottom row.**

**Table 1.** Information of the weather stations included in this study: elevation, z (m a.s.l.); code; data length, as initial and final dates of the time series; number of initially available daily records, $N_o$ (days); number of available daily records after the quality check, N (days); rate of days for cloudy, $N_{CI<0.3}$ (%), partially cloudy, $N_{0.3<CI<0.6}$ (%), and clear-sky conditions, $N_{CI>0.6}$, (%); and maximum, $R_{go\_max}$ (MJ m$^{-2}$ day$^{-1}$), mean, $R_{go\_mean}$ (MJ m$^{-2}$ day$^{-1}$), and minimum, $R_{go\_min}$ (MJ m$^{-2}$ day$^{-1}$), daily global radiation observed values. The selected descriptors for sky conditions and global radiation correspond to registered data after quality check.

| z | Code | Initial date | Final date | $N_o$ | N | $N_{CI<0.3}$ | $N_{0.6<CI<0.3}$ | $N_{CI>0.6}$ | $R_{go\_max}$ | $R_{go\_mean}$ | $R_{go\_min}$ |
|---|---|---|---|---|---|---|---|---|---|---|---|
| 781 | 602 | 26/01/2001 | 31/12/2018 | 6521 | 6370 | 8 | 23 | 69 | 33.80 | 18.49 | 0.80 |
| 942 | 608 | 01/02/2000 | 31/12/2018 | 6883 | 6686 | 6 | 26 | 68 | 34.20 | 18.83 | 0.70 |
| 950 | 601 | 05/09/2000 | 31/12/2018 | 6600 | 6449 | 7 | 27 | 66 | 33.00 | 18.17 | 0.60 |
| 975 | 853 | 21/11/2007 | 29/12/2018 | 2833 | 2827 | 8 | 30 | 62 | 32.37 | 18.01 | 1.00 |
| 1212 | 604 | 05/09/2000 | 31/12/2018 | 6665 | 6485 | 7 | 29 | 64 | 33.00 | 18.09 | 0.70 |
| 1332 | 803 | 27/08/2009 | 31/12/2018 | 3407 | 3282 | 7 | 22 | 71 | 33.41 | 18.95 | 0.71 |
| 1530 | 854 | 26/10/2007 | 16/12/2018 | 3176 | 3169 | 10 | 28 | 62 | 32.91 | 17.97 | 1.10 |
| 1732 | 857 | 16/11/2007 | 29/12/2018 | 3042 | 3034 | 11 | 25 | 64 | 32.84 | 18.31 | 0.81 |
| 1735 | 859 | 23/01/2008 | 21/11/2018 | 2577 | 2573 | 11 | 23 | 66 | 33.67 | 19.11 | 0.59 |
| 2141 | 804 | 10/10/2012 | 31/12/2018 | 2272 | 2206 | 7 | 21 | 72 | 33.91 | 19.05 | 0.82 |
| 2155 | 855 | 02/01/2008 | 30/11/2017 | 2522 | 2519 | 13 | 30 | 57 | 33.64 | 17.64 | 0.78 |
| 2300 | 858 | 09/03/2008 | 20/09/2017 | 2385 | 2380 | 12 | 28 | 60 | 34.58 | 17.99 | 0.99 |
| 2325 | 1002 | 15/11/2008 | 29/10/2012 | 951 | 951 | 8 | 22 | 70 | 35.60 | 20.47 | 1.55 |
| 2510 | 802 | 04/11/2004 | 31/12/2018 | 5050 | 4849 | 6 | 19 | 75 | 36.29 | 20.28 | 0.69 |
| 2867 | 1001 | 16/11/2007 | 01/01/2014 | 1071 | 1071 | 6 | 28 | 66 | 33.70 | 18.06 | 1.68 |
| 3097 | 860 | 23/01/2008 | 09/09/2018 | 1858 | 1705 | 13 | 25 | 62 | 35.79 | 18.20 | 1.12 |

## 3.3 Generation of global radiation maps

The GIS-based solar radiation model proposed by Aguilar et al. (2010) that was previously implemented and validated in a small subwatershed located in the southwest of Sierra Nevada (Fig. 1) was extended to the whole area in this study. For validation purposes, data registered at weather stations are considered to represent the average values of the 30 m cell of the DEM on which they are located (Batllés et al., 2008; Martínez-Durbán et al., 2009).

The main equations and flowchart of the model are shown in Appendix A. The complete explanation of the algorithms as well as the justification of the assumptions of the model can be found in detail in Aguilar et al. (2010).

The model was developed to be run using limited data but considering the agents that constitute the main sources of the spatial and temporal variability of solar radiation. Results generated by the model include hourly maps of diffuse, beam and reflected solar radiation values with minimum input data requirements as only topographic data, albedo estimations and measured daily global radiation records ($R_{go}$) at least at one weather station are required. As for the daily global radiation registers, even when

they are missing, their estimation from other more readily available meteorological data could always be a choice from the
literature (Hargreaves and Samani, 1982; Bristow and Campbell, 1984; Allen, 1997; Bechini et al., 2000; Winslow et al., 2001;
Donatelli et al., 2003, 2006; Yang and Koike, 2005; Diodato and Bellocchi, 2007; Wu et al., 2007; Ruiz-Arias et al., 2011; Liu
et al., 2012b; El Ouderni et al., 2013; Mullen et al., 2013).
The generation of global radiation maps with the model applied (Aguilar et al., 2010) requires a proper characterization of the
spatio-temporal patterns of albedo in the study site. 240 cloud-free Landsat imagery available for the study period from Landsat
5 TM (49 images), Landsat 7 ETM+ (141 images) and Landsat 8 OLI (50 images) with a 30 m spatial resolution were used.
Figure 3 shows the specific dates and sensors of the 240 images analyzed in this study. All images were first properly corrected,
and their reflectivity values computed (Pimentel et al., 2014). Albedo was then derived for each image following the same
procedure applied in Aguilar et al. (2010), which is based on the methodology described by Brest and Goward (1987), and
linearly interpolated on a daily time scale for the whole study period.

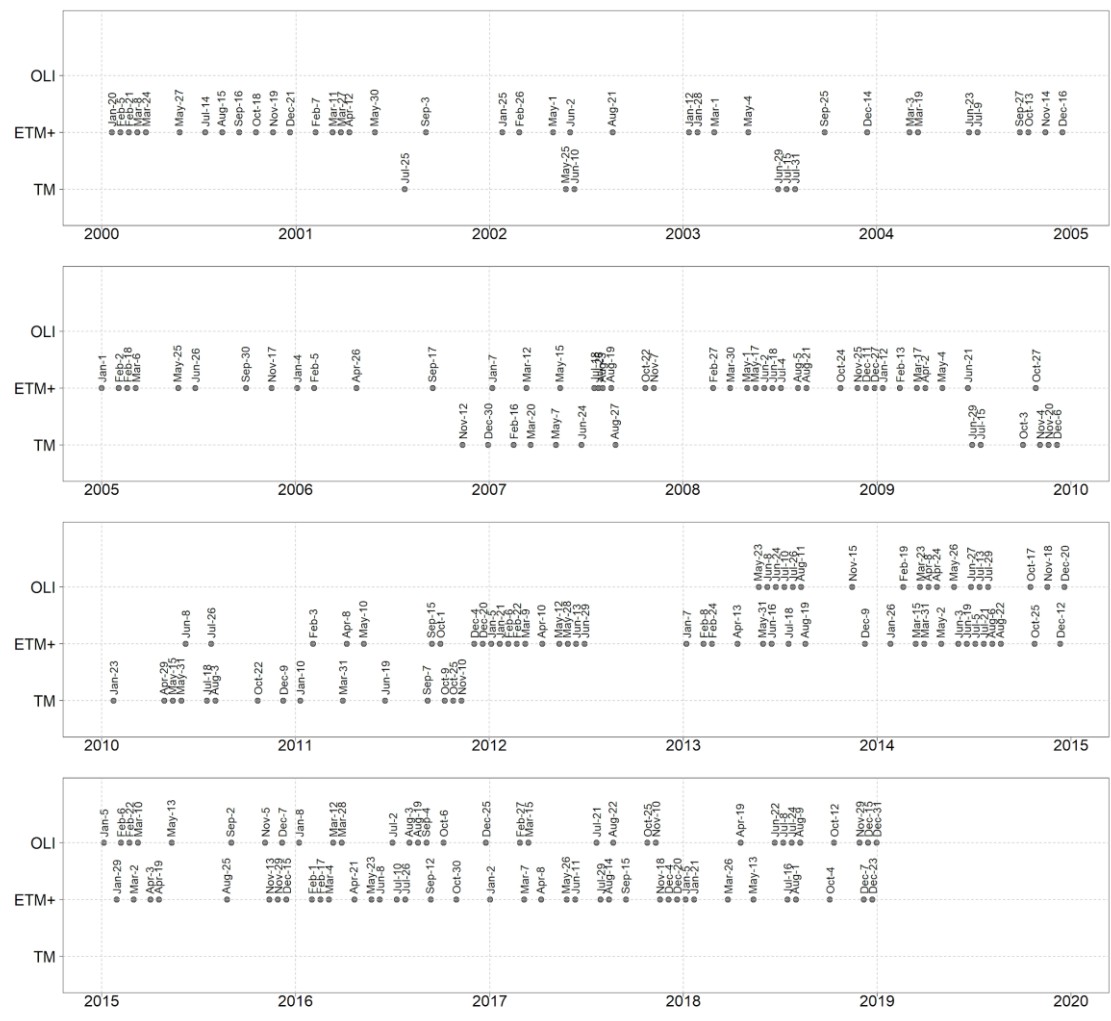

**Figure 3. Dates and sensors of each Landsat image analyzed in the study period (01 Feb 2000 - 31 Dec 2018).**

### 3.4 Cross-validation at weather stations

Once daily global radiation estimates were generated by the model a cross validation was applied at each weather station on the daily scale. This was carried out on a leave-one-out process, i.e., data from a weather station were removed from the input dataset to the model and predicted values ($R_{gp}$) at that weather station were then compared to observed data ($R_{go}$).

Different indicators were computed to quantitatively evaluate the performance of the model (Muneer et al., 2007):

-The Root Mean Square Error (RMSE) (Eq. 1), where $R_{gp}$ and $R_{go}$ are the predicted and observed daily global radiation (MJ m$^{-2}$ day$^{-1}$), respectively, and N is the number of observed daily data. It measures the difference between values predicted by the model and those which were actually observed.

$$RMSE = \sqrt{\frac{\sum \left(R_{gp} - R_{go}\right)^2}{N}}$$

(1)

-The deviation from the 1:1 line of observed vs. predicted daily solar radiation values. Linear fits forced through the origin were obtained (Eq. 2) and the slopes ($\alpha$ in Eq. 2) are desired to be equal to 1. The coefficient of determination, $R^2$, as the ratio of the explained variation to the total variation, was also computed.

$$R_{gp} = \alpha \cdot R_{go}$$

(2)

The RMSE values and linear fits were obtained for the whole dataset at each weather station, and also for different cloudiness levels to consider different atmospheric states that may condition the performance of the model according to previous studies (Batllés et al., 2008; Martínez-Durbán et al., 2009; Ruiz-Arias et al., 2009). Based on the cloudiness three types of weather conditions were analyzed: cloudy days (CI<0.3), partly cloudy days (0.3≤CI<0.6) and clear-sky days or cloudless days (CI≥0.6).

The cross-validation assessment is summarized in Figure 4. With the global datasets (in black in Fig. 4), a very close approximation of the model estimates to recorded data was obtained (mean $\alpha$ value of 0.98 and mean $R^2$ values of 0.91). RMSE values varied for the different stations and ranged from 1.81 (station 804) to 3.76 (station 860) with a mean value of 2.63 MJ m$^{-2}$ day$^{-1}$.

When the analysis was carried out in terms of the cloudiness level, a general overestimation by the model (e.g., a mean $\alpha$ value of 1.41) was always seen on cloudy days (CI≤0.3). In contrast, on clear-sky days (CI>0.6) slopes were very close to 1 with a mean $\alpha$ value of 0.96. An intermediate behavior was found on partly cloudy days (0.3<CI≤0.6) when the model slightly under predicted (e.g., stations 854 and 608) or over predicted depending on the weather station. As for RMSE values, the lowest values were always found for clear sky days, when the cloud influence is minimal and the attenuation is mostly explained by changes in the atmospheric transmittance, followed by partly cloudy days with mean values of 2.07 and 3.07 MJ m$^{-2}$ day$^{-1}$, respectively. The highest RMSE values were always found on cloudy days with mean values of 3.70 MJ m$^{-2}$ day$^{-1}$. The high proportion of clear-sky days (65%) and the low RMSE values on these days (2.07 MJ m$^{-2}$ day$^{-1}$) revealed the general good agreement of the model estimates with observed data. This is especially important in semiarid environments, where energy-

limited hydrological processes (e.g., soil moisture depletion, evaporation, or snowmelt) are more relevant on clear-sky days
and they must be carefully computed in water and energy balance modeling, irrigation scheduling, etc. (Chen et al., 1999;
Mamassis et al., 2012).
There is no clear pattern in the errors obtained with the elevation of the stations. The goodness of the model estimates was
more affected by the interaction of the different characteristics of the weather station (e.g., slope, aspect, surrounding terrain
configuration, orographic effects in the vertical development of clouds, etc.) than by the height of the station itself.
To validate the modeling scheme applied, another well-known GIS-based solar radiation model, Solar Analyst (SA) (Fu and
Rich, 2000b) was also applied in the study site. Error values in the approximation to observed data and linear fits obtained in
SN are shown in Appendix B. In view of the errors obtained with SA estimates (Table B.1) we can select the modeling scheme
here proposed (Subsection 3.3) over SA to analyze the spatial and temporal behavior of solar radiation in SN.
The errors obtained in Figure 4 were within the order of magnitude of those found in previous studies in other mountainous
areas (Yang et al., 2006; 2010; Zhang et al., 2020) and slightly improved those previously obtained on a small subarea (10 x 5
km$^2$) in the north-eastern side of SN. Here, Tovar-Pescador et al. (2006) analyzed the application of SA in clear sky days with
a 168 global radiation dataset from 14 weather stations located at between 1091 and 1659 m.a.s.l. They obtained $R^2$ values of
0.75, similar to the value here obtained with SA estimates in the whole SN area (0.77 in Table B.1) but lower than the $R^2$ equal
to 0.99 obtained with the model (in orange in Figure 3). Then, Batllés et al. (2008) in another application of SA in the same
area with a 2-year daily dataset obtained the best performances for clear-sky days. RMSE values obtained in clear-sky days in
the present study, of 11.1 % (2.07 MJ m$^{-2}$ day$^{-1}$); were the same as those obtained by Batllés et al. (2008) for clear sky days
(11%). Later, Ruiz-Arias et al. (2009) evaluated the application of four different GIS-based solar radiation models with a 523
global radiation dataset at the same study site. RMSE values for the global dataset ranged between 1.99 and 7.28 MJ m$^{-2}$ day$^{-1}$
$^1$ depending on the model.
The order of magnitude of the errors (Figure 4) and its comparison with those obtained with more computationally and data
demanding GIS-based models in previous studies let us to conclude that the model is the best choice to generate global radiation
data series in SN.
Therefore, once the model was validated in the study site, daily $R_g$ maps were generated and aggregated at the monthly and
annual scales.

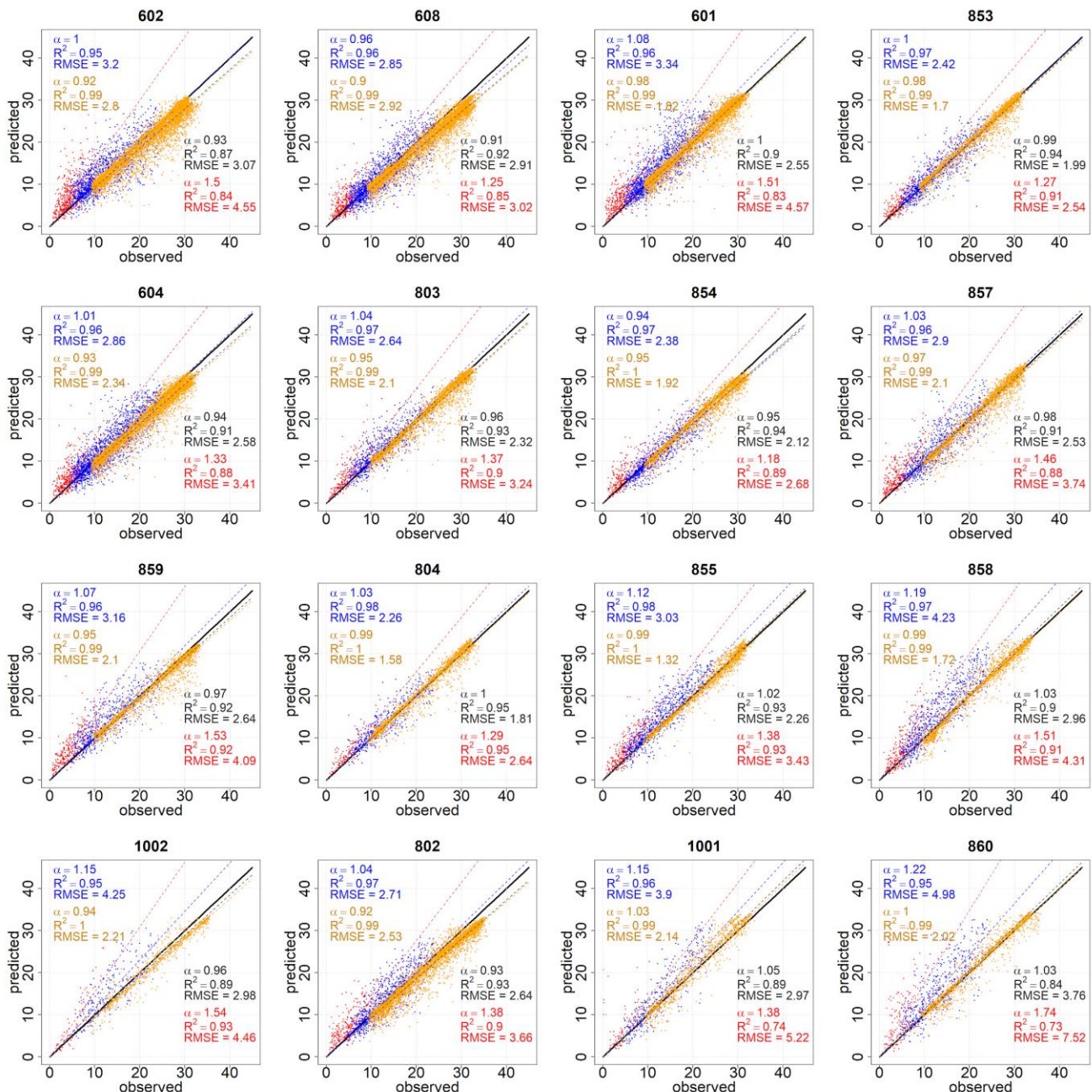

Figure 4. Cross validation analysis. Linear fits of daily predicted *vs.* observed $R_g$ (MJ m$^{-2}$ day$^{-1}$) at each of the selected stations for the global data (black), cloudy (CI<0.3 - red), partly cloudy (0.3<CI<0.6 - blue) and clear-sky days (CI>0.6 - orange). Stations are sorted by increasing altitude from left to right and from the top to the bottom row.

## 4 Results

Daily, monthly, and annual $R_g$ datasets in SN are analyzed in this section at two spatial scales. First, the results at the weather station scale are presented. Thus, possible relationships between altitude and/or location of the weather station with the different $R_g$ statistics and how this relation changes with the temporal scale of analysis can be assessed. Then, the $R_g$ maps that can be downloaded as specified in section 6 are analyzed.

### 4.1 Daily time series of global radiation in Sierra Nevada

Figure 5 shows the statistical distribution of the daily $R_g$ at each weather station ordered by increasing altitude and illustrates several questions. First, there is a very similar interquartile range among stations. Second, there are greater variations in the maximum daily $R_g$ among the different stations with a mean value of 34.0 MJ m$^{-2}$ day$^{-1}$. Third, even though a slight increase with altitude can be shown in the high extreme statistics of the daily $R_g$ values (e.g., in the maximum or in the 90$^{th}$ percentile), there is not a clear trend. Therefore, other factors such as orientation, proximity to the sea or the terrain configuration in the surrounding terrain as suggested by Batllés et al. (2008) constitute relevant features in the study site.

Figure 6 shows an example of the spatial distribution in three representative days of cloudy, partially cloudy, and clear-sky conditions. Here the spatial distribution is clearly influenced by the topography of SN, especially in the clear sky day.

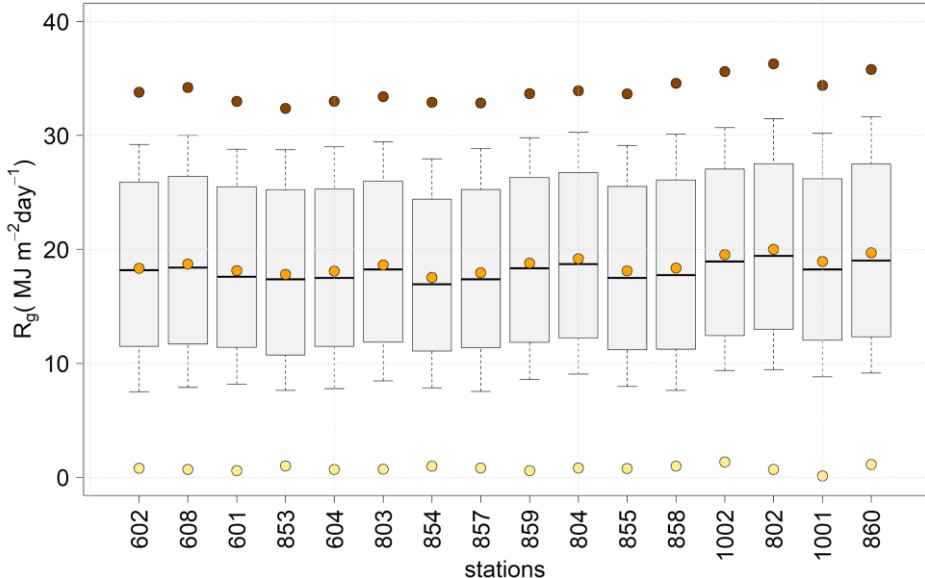

**Figure 5. Statistical distribution of daily $R_g$ (MJ m$^{-2}$ day$^{-1}$) time series at each of the selected stations over the study area. The box shows 50% of the data, delimited by Q1 (lower) and Q3 (upper), the solid line represents the median, and whiskers show 10$^{th}$ and 90$^{th}$ percentiles. Brown, orange and yellow dots represent daily maximum, mean and minimum time series values. Stations are sorted by increasing altitude from left to right.**

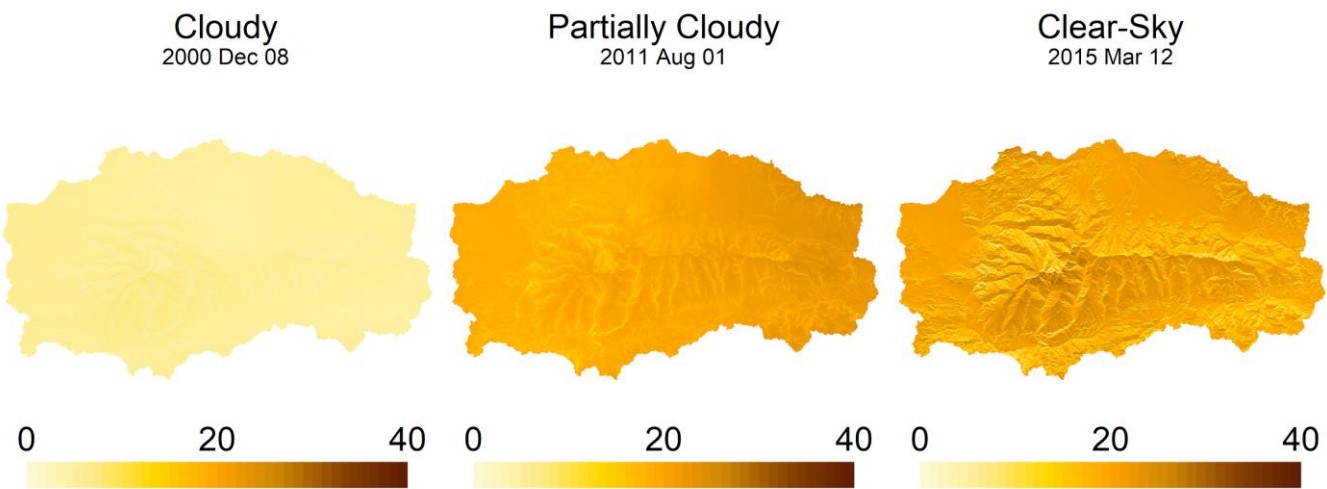

284

**Figure 6. Daily $R_g$ (MJ m$^{-2}$ day$^{-1}$) in SN for three selected days that represent the three levels of cloudiness considered in this study: cloudy, partially cloudy and clear-sky.**

287

**4.2 Monthly time series of global radiation in Sierra Nevada**

The statistical distribution of monthly $R_g$ per weather station (Fig. 7) shows that in every station: i) July and December constitute the months with the highest and the lowest values of $R_g$, respectively; ii) there is a quite linear increase in the monthly $R_g$ values from January to July and a sudden drop in August with a slightly convex evolution till December; and iii) the interquartile range is significantly higher in the spring and fall, than in the summer and winter months.

The increase in the high extreme statistics of radiation with the altitude of the weather station becomes more apparent at the monthly scale (Fig. 7) than at the daily scale (Fig. 5) previously analyzed. Thus, maximum values of around 1000 MJ m$^{-2}$ month$^{-1}$ are reached in July in the highest stations (e.g. 1002, 802, 1001 and 860 in Fig. 7) whereas this value decreases to around 910 MJ m$^{-2}$ month$^{-1}$ in the four lowest stations with the exception of station 608.

Monthly $R_g$ maps show significant spatial differences of up to 200 MJ m$^{-2}$ month$^{-1}$ in both the mean monthly values (Fig. 8) that clearly follow the terrain configuration with summits and valleys receiving high and low solar radiation values, respectively. For example, the area in the north of SN that is highly shadowed by the highest peaks in the Iberian Peninsula (Mulhacen and Veleta with 3482 and 3396 m a.s.l., respectively) is easily visible, with the lowest relative levels of insolation received within SN especially in the summer months (June, July and August in Fig. 8).

Both, maps of the monthly mean and standard deviation of $R_g$ (Fig. 8) and the statistical distribution of the monthly $R_g$ in the study site (Fig. 9), show the same behaviour as the one obtained at the weather stations regarding: i) July and December as the months with the highest and lowest values of $R_g$ received in SN; and ii) the highest scatter in the monthly $R_g$ values in the spring and fall months.

306

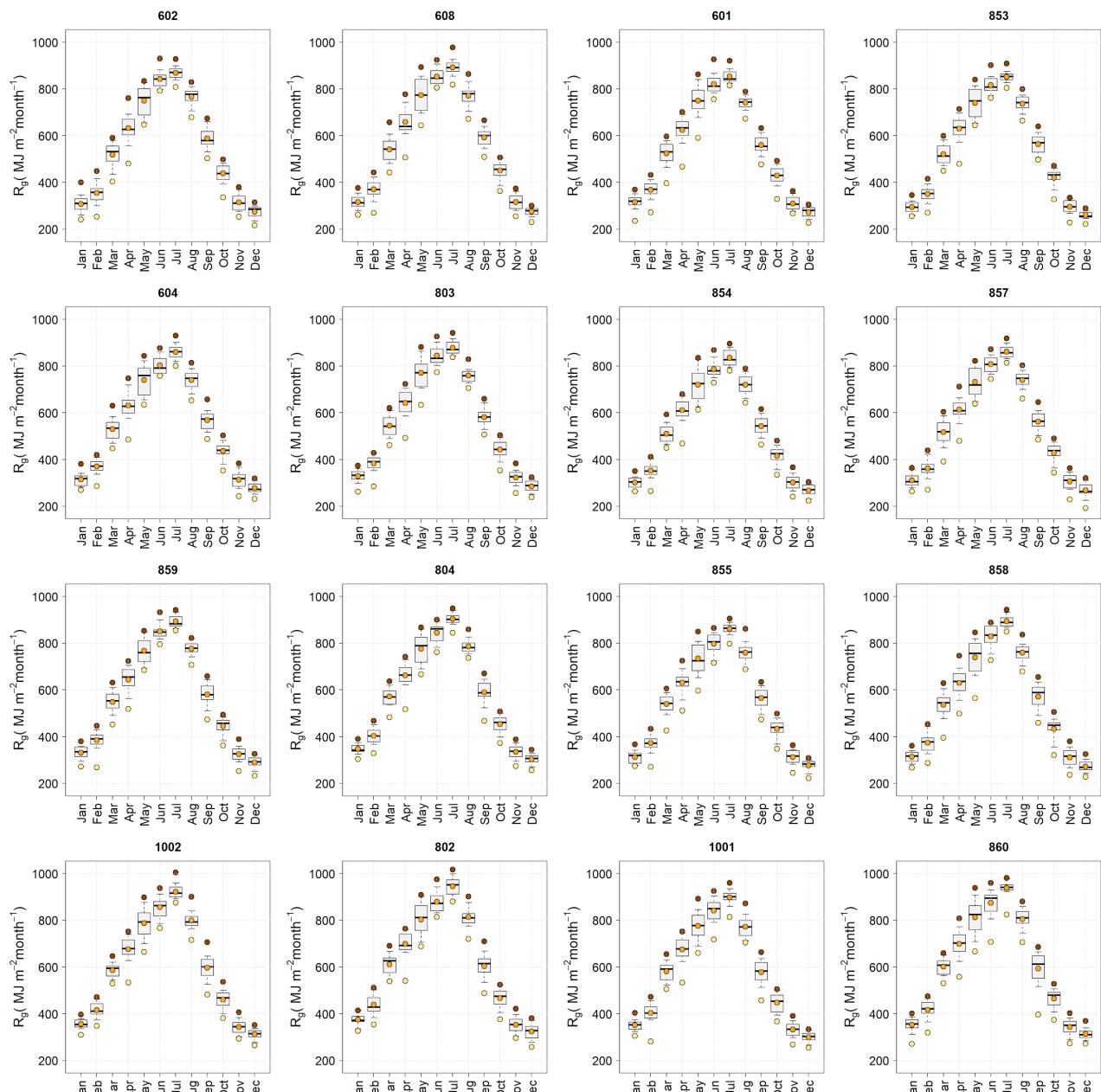

307

Figure 7. Statistical distribution of monthly $R_g$ (MJ m$^{-2}$ month$^{-1}$) time series at each of the selected stations over the study area. The box shows 50% of the data, delimited by Q1 (lower) and Q3 (upper), the solid line represents the median, and whiskers show 10[th] and 90[th] percentiles. Brown, orange and yellow dots represent monthly maximum, mean and minimum time series values. Stations are sorted by increasing altitude from left to right and from the top to the bottom row.



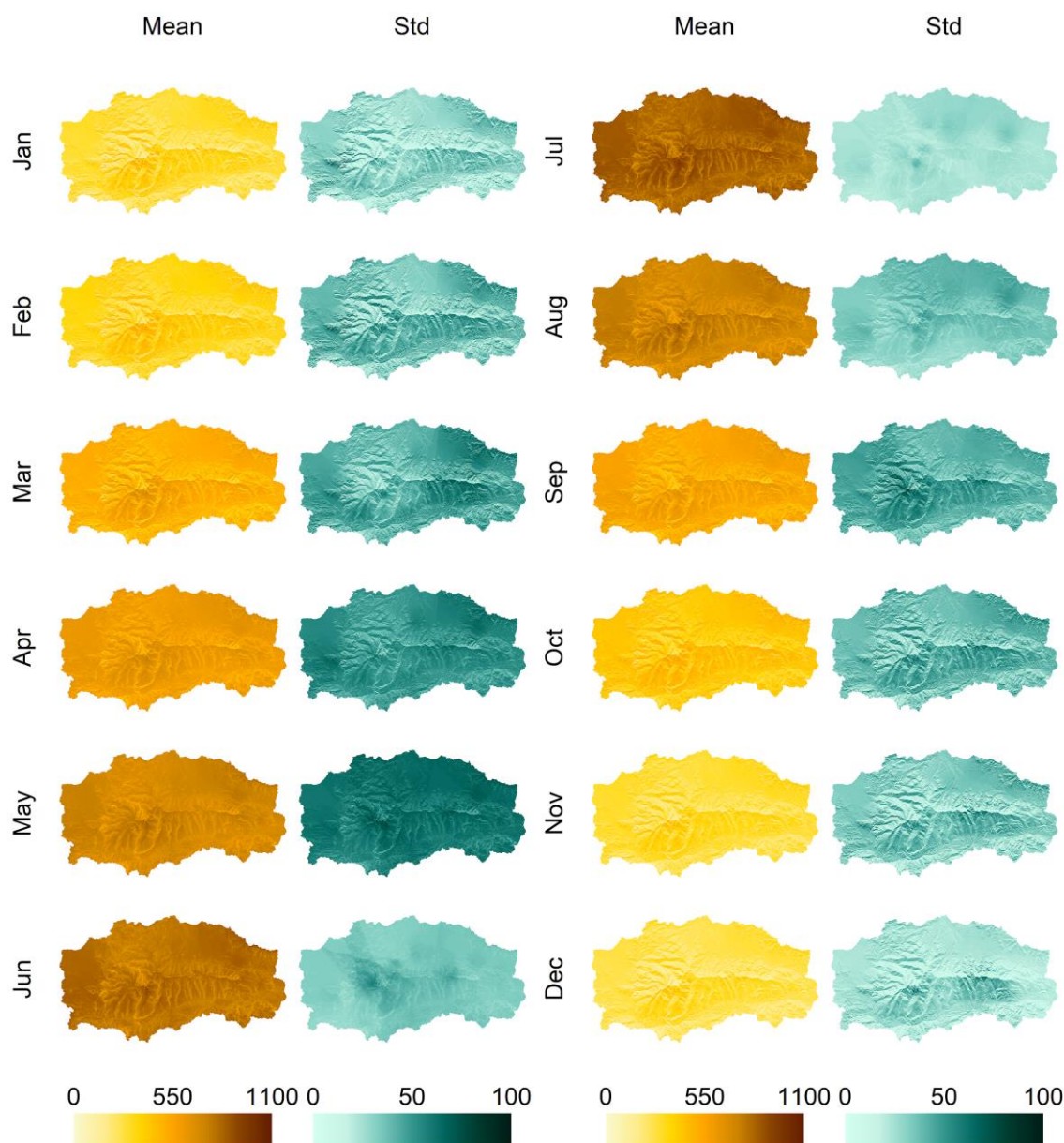


Figure 8. Monthly average and standard deviation of $R_g$ (MJ m$^{-2}$ month$^{-1}$) in the study period (2000-2018) in SN.


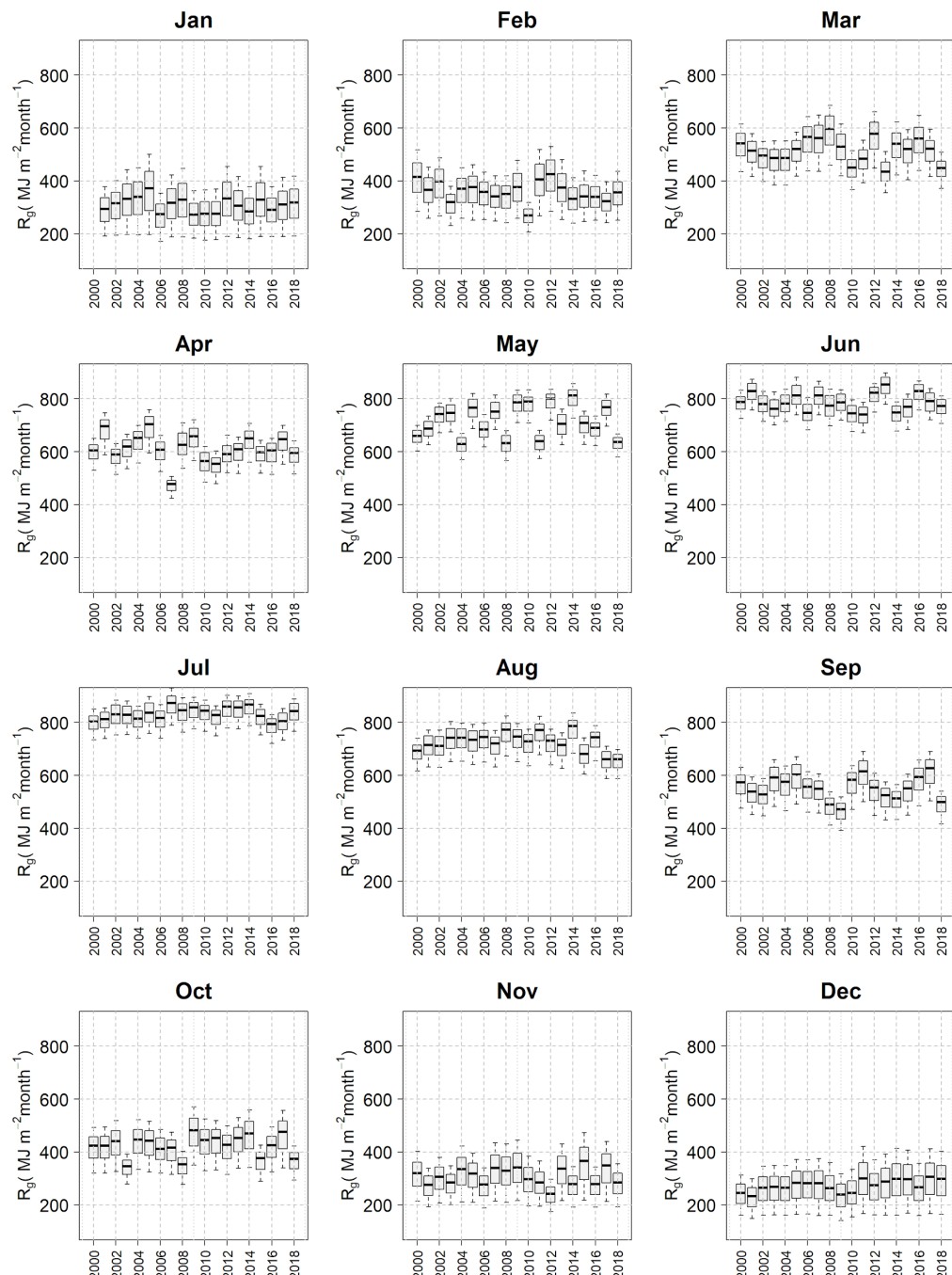


Figure 9. Statistical distribution of the monthly $R_g$ (MJ m$^{-2}$ month$^{-1}$) values throughout the study area. Whisker boxes represent the 10th, 25th, 50th, 75th and 90th percentiles of each monthly map per year


For the study period (2000-2018), there is a great heterogeneity in the statistical distribution of the monthly $R_g$ in the study site
(Fig. 9) especially in the incoming radiation along the months of the wet season (October-May). In this way, in the most
insolated years in the study period (2005 and 2012), significantly higher monthly radiation values were found in certain months
of the springtime (March and May 2012 and April 2005). In those months, the higher than usual rate of clear-sky over cloudy
days finally determines the annual differences in the incoming global radiation in SN.
When considering the temporal evolution of the distribution of $R_g$ within the monthly maps in SN (Fig. 10), certain interannual
differences can be observed along the study period, such as the existence of certain months in spring with unexpected low
monthly radiation values (eg. 2001, 2004, 2007 and 2008), or two relative maximum monthly $R_g$ values (e.g. 2009, 2010 and
2014). Moreover, Figure 10 shows a higher scatter in the monthly maximum (June-August) and minimum (November-January)
$R_g$ values in SN than when the analysis is carried out at each weather station (Fig. 7).

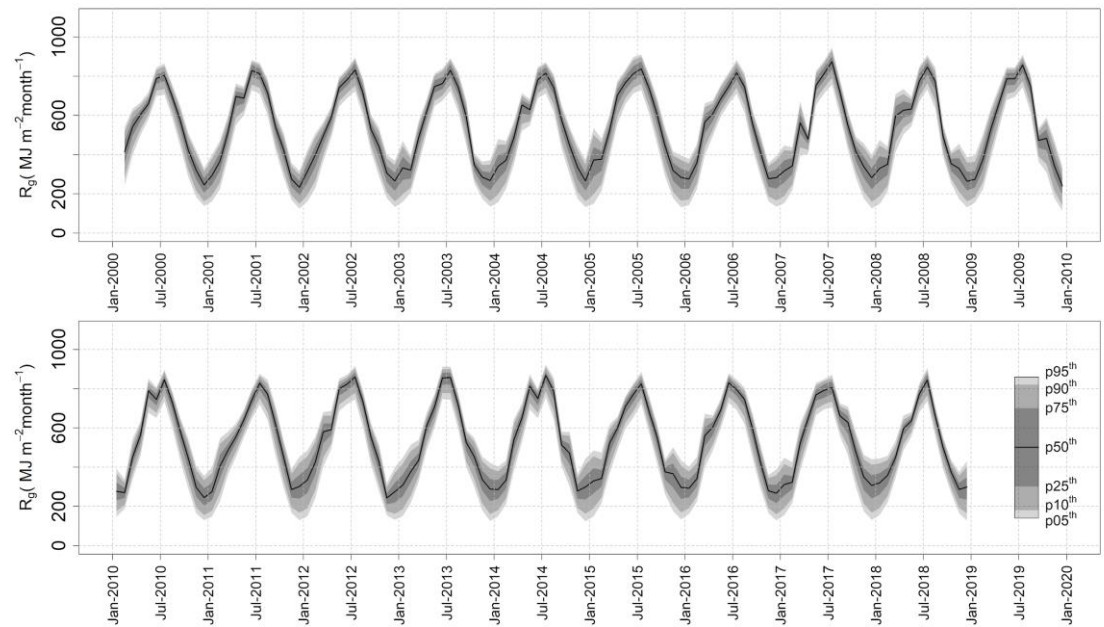


**Figure 10. Evolution of the statistical distribution of monthly $R_g$ (MJ m$^{-2}$ month$^{-1}$) in the study period (2001-2018) throughout the study area. Grayscale colours represent the following percentiles: 5th, 10th, 25th, 50th, 75th, 90th and 95th.**

## 4.3 Annual times series of global radiation in Sierra Nevada

Unlike at the daily scale (Fig. 5), a great variability among the different weather stations in terms of the global radiation
received at the annual temporal scale is found (Fig. 11). Thus, we find minimum annual $R_g$ values from 5920 MJ m$^{-2}$ year$^{-1}$ in
station 854 to around 6750 MJ m$^{-2}$ year$^{-1}$ in station 1002. This difference is even bigger in the maximum annual $R_g$ values from
6700 to 7720 MJ m$^{-2}$ year$^{-1}$ in stations 854 and 802, respectively, and is also shown in the interquartile range.
When analyzing the influence of altitude, the weather stations above 1500 m a.s.l (854, 857, 859, 804, 855, 858, 1002, 802,
1001, 860 in Fig. 11) show their altitudinal gradient in all the statistics of the annual $R_g$ values considered.
Annual $R_g$ maps (Fig. 12) show the same spatial differences that follow the terrain configuration as those observed in the
monthly time series (Fig. 8). For example, the area in the north of SN that is highly shadowed as previously mentioned
corresponds to the area with the mean minimum annual values received in the study period, 4063 MJ m$^{-2}$ year$^{-1}$, that only
represents 63% the mean annual accumulated values in SN (6316 MJ m$^{-2}$ year$^{-1}$).

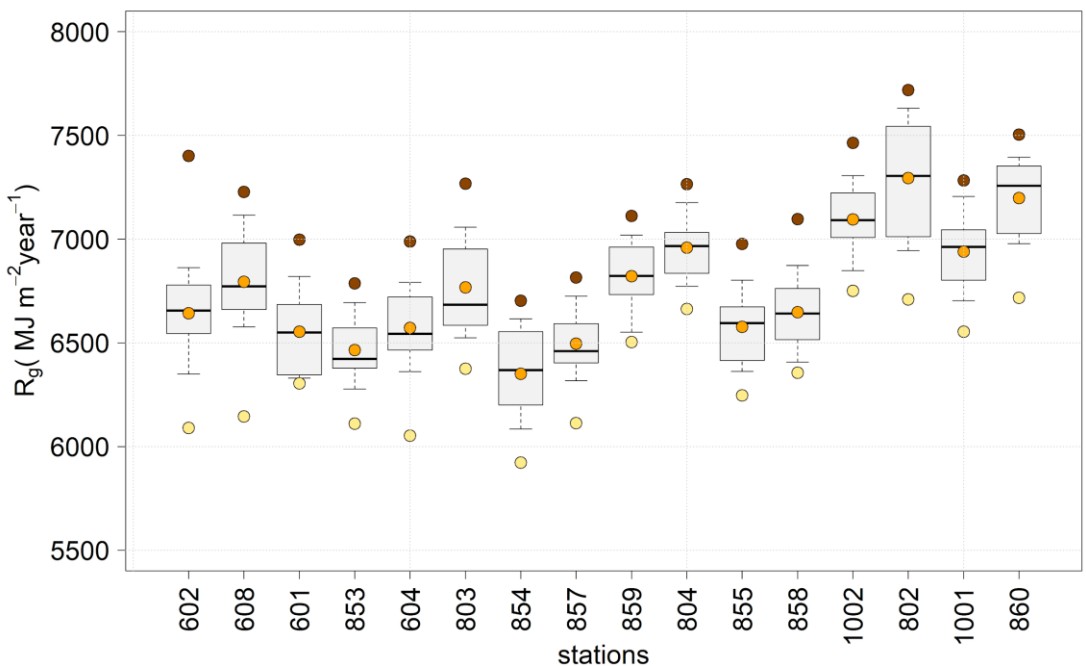


**Figure 11. Statistical distribution of annual $R_g$ (MJ m$^{-2}$ year$^{-1}$) time series at each of the selected stations over the study area. The**
**box shows 50% of the data, delimited by Q1 (lower) and Q3 (upper), the solid line represents the median, and whiskers show 10th**
**and 90th percentiles. Brown, orange and yellow dots represent annual maximum, mean and minimum time series value. Stations are**
**sorted by increasing altitude from left to right.**


Significant interannual differences can be easily shown with differences in the mean annual $R_g$ value in the study area of up to
800 MJ m$^{-2}$ year$^{-1}$ between 2005 and 2018. Such years with particularly high and low annual incoming radiation also presented
higher (6800 MJ m$^{-2}$ year$^{-1}$) and lower median annual $R_g$ values (6200 MJ m$^{-2}$ year$^{-1}$), respectively, than the annual median for
the whole study period in SN (6456 MJ m$^{-2}$ year$^{-1}$) (Fig. 13). These results agree with the annual irradiation map obtained by
Batllés et al. (2008) in the north-eastern part of SN. They reported maximum and minimum annual values of 7516 and 2342
MJ m$^{-2}$ year$^{-1}$ on the summits and in deep valleys, respectively, and thus, concluded that irradiation levels were more related
to topographic characteristics than to altitude.

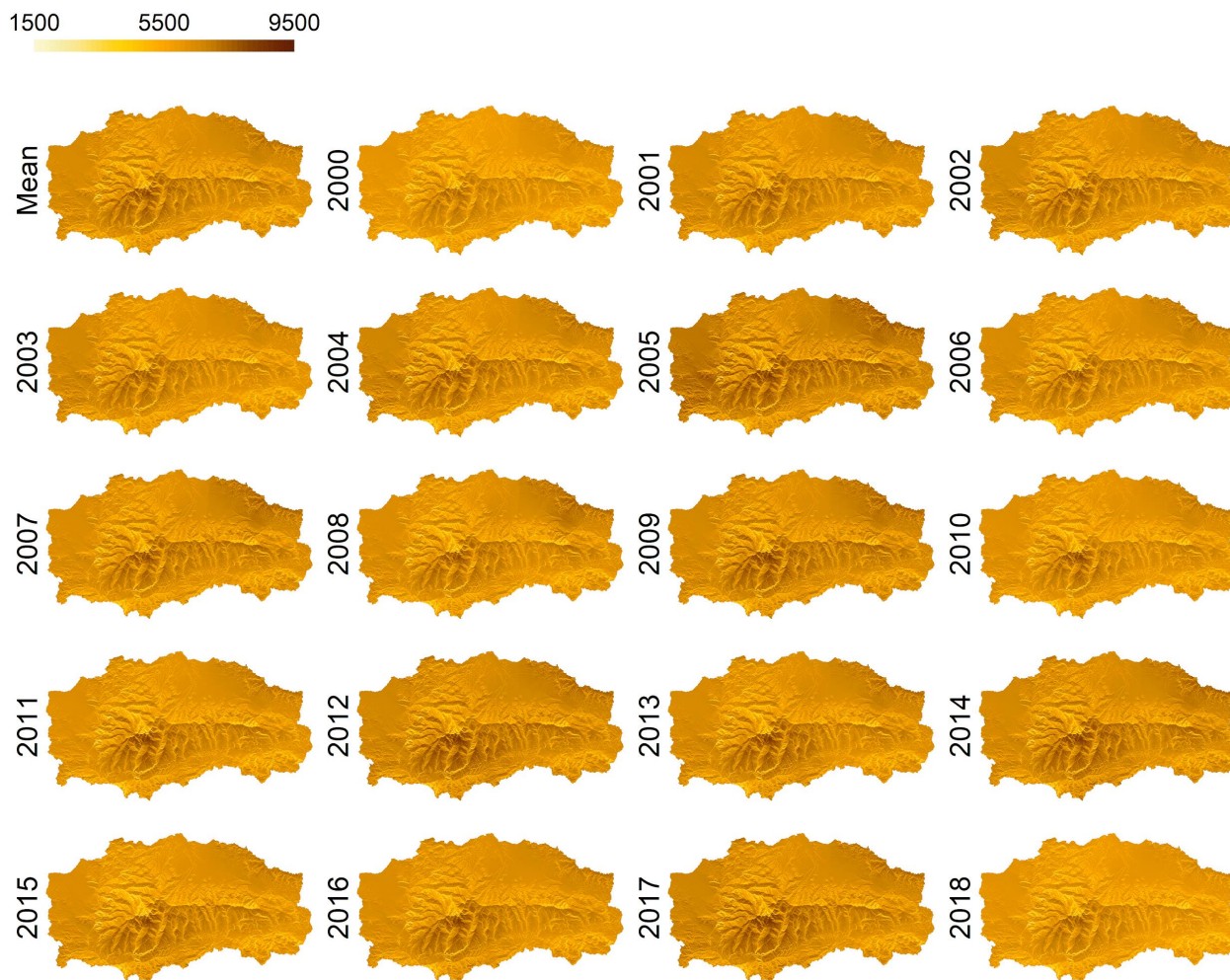

Figure 12. Annual global radiation (MJ m$^{-2}$ year$^{-1}$) in the study period (2001-2018) in SN.

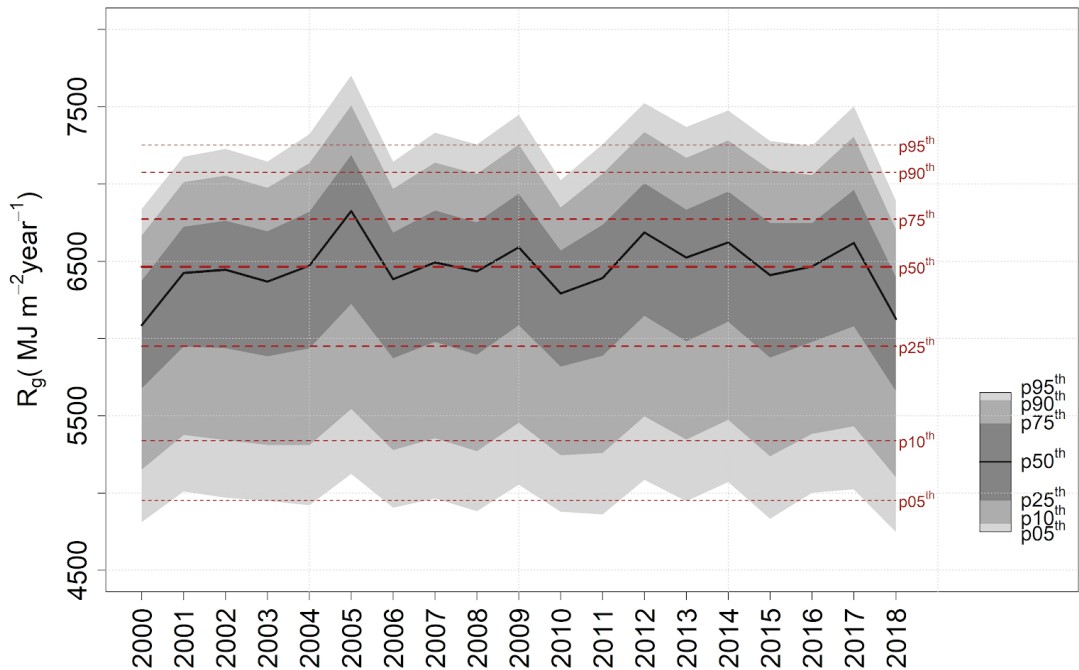


**Figure 13. Evolution of the statistical distribution of annual R$_g$ (MJ m$^{-2}$ year$^{-1}$) in the study period (2001-2018) throughout the study area. Dashed lines represent the mean values of the percentiles analyzed.**

## 5 Data availability

The daily, monthly and annual global radiation maps derived in this study can be accessed and downloaded in.ncdf format from: https://doi.pangaea.de/10.1594/PANGAEA.921012 (Aguilar et al., 2021). Besides, a .txt file containing the availability (code 1) or gaps (code 0) in the daily R$_g$ observations at each weather station has been added as a supplement to this paper. Hourly datasets were also computed in this study but due to their large storing capacity requirements they have not been included in the data repository specified above. Thus, hourly maps can be provided for certain dates upon request to the authors. However, a validation of these hourly datasets like the one applied in the daily estimates at the weather stations has not been specifically carried out in this study. Therefore, in case hourly maps are requested to the authors, these data should be taken with caution as the only available validation in SN was carried out at one weather station (802 in Fig. 1) and for a shorter period (2004-2010) in Aguilar et al. (2010).

## 6 Final remarks

This study presents nineteen years (2000-2018) of daily, monthly, and annual global radiation maps of high spatial resolution (30 m) in a high mountain Mediterranean site. In these areas the common lack of weather stations in high altitudes makes it difficult to accurately assess solar radiation spatial patterns.

A GIS-based modelling scheme based on measurements or estimations of incoming daily global radiation was applied and validated in the sixteen weather stations available at this unique study site. Mean RMSE values ranged from 1.81 to 3.76 MJ $m^{-2}$ $day^{-1}$, depending on the weather station. The best estimations were always obtained on clear-sky days, when mean RMSE values decreased to 2.07 MJ $m^{-2}$ $day^{-1}$. The largest errors were obtained on cloudy days, which constitute on average 10% of the daily datasets, and, therefore, future research should be conducted in order to improve the estimations in these situations keeping the minimum input data requirements (daily global radiation data) advantage of the model. However, the high proportion (65%) of clear-sky days, and the low RMSE values on those days, allow one to conclude that there is a good agreement between the model estimates and observed data in the study site.

Spatial differences of around 2000 MJ $m^{-2}$ $yr^{-1}$ were found within each year analyzed. In addition, significant differences were easily shown between the years in mean incoming values of up to 800 MJ $m^{-2}$ $yr^{-1}$. Those differences were mostly due to the variability in the incoming radiation at the wet season (October-May), with higher rates of clear-sky days in the most insolated years (e.g., 2005).

Thus, we can affirm that the modeling scheme here applied is an efficient option in semiarid mountainous areas, where daily global radiation datasets constitute the only source of solar radiation data.

Time series of these surface global radiation datasets can be used to analyze inter-annual and seasonal variation characteristics of the global radiation received in SN with high spatial detail (30 m). The availability of long global radiation datasets allows to capture the annual variability within each cycle of the Sun activity, as reported in the literature (Scaffetta and Wilson, 2013), and thus estimate its contribution to the annual variability of other climate variables in these semiarid mountainous areas.

Dense and properly-maintained weather station networks in mountainous areas are rarely available. Thus, these datasets can also be used as cross-validation reference data for other global radiation distributed datasets generated in SN with different spatio-temporal interpolation techniques.

These results can also assess the order of magnitude of different sources of spatial variability (altitude/slope/aspect gradients) as well as the seasonal range of variation at different time scales and their annual variability. This estimation may provide a first estimate of the order of magnitude of uncertainty of average calculations or spatial interpolation from a scarce number of weather stations in Mediterranean and semiarid mountain areas.

The correct assessment of the solar radiation regime is crucial to correctly determine the temporal evolution of energy-limited hydrological processes such as the snow layer dynamics, soil moisture depletion and evapotranspiration (Tomas-Burguera et al., 2019). Thus, as a key input parameter for the water and energy balance, these high spatial resolution solar radiation time series are useful not only for research on the snow domain and water planning in SN in the application of hydrological

modelling, but in many other applications. For example, within the agricultural sector in the estimations of evapotranspiration
for irrigation scheduling, ecology and biodiversity studies, stand-alone solar energy facilities designing and location,
recreational activities in the area that strongly rely on the hydro-meteorological conditions of SN, etc.. Finally, this work
contributes to feed research related to some key questions in hydrology, as UPH 16 and UPH 5 identified by Blöschl et al.
413  (2019).


**Author contributions**
CA, in collaboration with MJP, conceived the research. CA processed the data, applied the quality control to the raw global
radiation data, modelled global radiation datasets and developed the cross-validation algorithms. RP processed satellite data,
generated albedo maps for the study period, prepared the final figures and the available datasets generated in the study. CA
prepared the manuscript with contributions of MJP and RP; all authors discussed and revised the final text.

**Competing interests**
The authors declare that they have no conflict of interest.

**Acknowledgements**
This study was supported by the following research projects funded by Spanish Ministry of Science and Innovation - MICINN:
Research Project RTI2018-099043-B-I00, "Operability in hydrological management under snow torrentiality/drought
conditions in high mountain in semiarid watersheds"; and, by Spanish Ministry of Economy and Competitiveness - MINECO:
Research Project CGL 2014-58508R, "Global monitoring system for snow areas in Mediterranean regions: trends analysis and
implications for water resource management in Sierra Nevada", and Research Project CGL 2011-25632, "Snow dynamics in
Mediterranean regions and its modelling at different scales. Implication for water management". Moreover, the present work
was partially developed within the framework of the Panta Rhei Research Initiative of the International Association of
Hydrological Sciences (IAHS) (Working Groups Water and energy fluxes in a changing environment and Mountain
Hydrology). Rafael Pimentel acknowledges fundings by the modality 5.2 of the *Programa Propio-2018* of the University of
Cordoba and the *Juan de la Cierva Incorporación* Programme of the Ministry of Science and Innovation (IJC2018-038093-I).
The continuous support of the Natural and National Park of Sierra Nevada has also been determinant for the development of
this line of research since 2002. Finally, tremendous appreciation is extended to all the weather station networks that maintain
and make accessible datasets to scientific research.

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

## Appendix A. Solar radiation equations

The sequence followed by the model is summarized in Figure A1. Computations are classified at the point scale of weather
stations (Point) and the distributed scale of grids of the Digital Elevation Model (DEM) (Distributed). The complete
explanation of the algorithms and assumptions of the model can be found in detail in Aguilar et al. (2010).
Firstly, daily extraterrestrial radiation ($R_{ext}$ in MJ m$^{-2}$ day$^{-1}$) is computed by integrating the extraterrestrial radiation incident
upon a horizontal surface relative to the sun's beams from sunrise to sunset (Eq. A1).

$$R_{ext} = E_o \cdot I_{SC} \cdot \cos(\theta_z) \tag{A1}$$

where $I_{SC}$ is the solar constant (1367 W m$^{-2}$), $\theta_z$ is the zenith angle and $E_o$, the eccentricity factor. These variables were
computed following the equations in Dozier et al. (1981).
Then, the daily clearness index (CI), as the ratio of observed daily global radiation ($R_{go}$ in MJ m$^{-2}$ day$^{-1}$) to the daily
extraterrestrial radiation, is computed at each weather station (Eq. A2).

$$CI = \frac{R_{go}}{R_{ext}} \tag{A2}$$

CI is expressed in terms of two factors, $CI_{CS}$ and $fCI_{cl}$. The first term represents the influence of atmosphere under clear-sky
conditions over solar radiation, while the second term includes the cloudiness effects that decrease the final incoming solar
radiation (Eq. A3). The approximation of Ineichen and Perez (2002) is used to compute the global radiation under clear-sky
conditions, $R_{gcs}$, and thus, distributed hourly $R_{gcs}$ values are obtained from the sun elevation angle, the height of the cell, the
Linke turbidity factor ($T_L$) and the atmospheric mass obtained following the parameterization of Kasten and Young (1989).
Thus, hourly $CI_{CS}$ values can be computed cell by cell and then the mean daily distributed values are generated. Once daily CI
and $CI_{CS}$ values are known, $fCI_{cl}$ is obtained at each weather station from Eq. A3 and spatially interpolated following the
inverse distance weighted (IDW) method. From daily $CI_{CS}$ and $fCI_{cl}$ maps, daily interpolated CI and $R_g$ values can be obtained
at cell scale from Eq. A3 and A4.
$$CI = CI_{CS} \cdot fCI_{cl} \tag{A3}$$

$$R_g = R_{ext} \cdot CI_{CS} \cdot fCI_{cl} \tag{A4}$$

Topographic effects need to be evaluated for the different sun positions during the day and thus, hourly values of the different
components need to be derived. Two different procedures are currently available in the model. The first one proposed in
Aguilar et al. (2010) applies Jacovides et al. (1996) (Eq. A5.1) to produce the daily diffuse ($R_d$ in MJ m$^{-2}$ day$^{-1}$) and daily beam
values ($R_b$ in MJ m$^{-2}$ day$^{-1}$). The model finally computes hourly beam and diffuse values on horizontal surfaces ($r_b$ and $r_d$, both
in MJ m$^{-2}$ h$^{-1}$), from the daily amounts and following the temporal pattern of extraterrestrial hourly radiation during the day.
$$\frac{R_d}{R_g} = \begin{cases} 0.992 - 0.0486CI & CI \leq 0.1 \\ 0.954 + 0.734CI - 3.806CI^2 + 1.703CI^3 & 0.1 < CI \leq 0.71 \\ 0.165 & CI > 0.71 \end{cases} \tag{A5.1}$$

The second approach uses the temporal pattern of extraterrestrial hourly radiation, $r_{ext}$, to generate hourly global values, $r_g$
according to previous studies (Chen et al., 1999; Ruiz-Arias et al., 2011). Then, the hourly regressive model developed by
Ruiz-Arias et al. (2010) is applied to estimate the hourly diffuse values (Eq. A5.2) from the hourly CI, $CI_h$, as the ratio of $r_g$ to
$r_{ext}$. This model was implemented as it has been validated over Europe and USA using ground data from different sites,
including some Spanish stations (Ruiz-Arias et al., 2010). Hourly beam values ($r_b$) are thus obtained on a cell basis as the
difference between global and diffuse hourly radiation distributed values.
$$\frac{r_d}{r_g} = 0.952 - 1.041e^{-\exp(2.3 - 4.702 \cdot CI_h)} \tag{A5.2}$$

First applications at the study site have shown negligible differences between both partitioning schemes. The differences with
daily recorded data were insignificant in the second decimal place of error values. Thus, the results presented in this study
were obtained with the original scheme of Aguilar et al. (2010) (Eq. A5.1) while the authors continue working on the
improvement on the partitioning scheme of the model.
Then, the topographic correction is carried out and depending on the component, different procedures are applied.

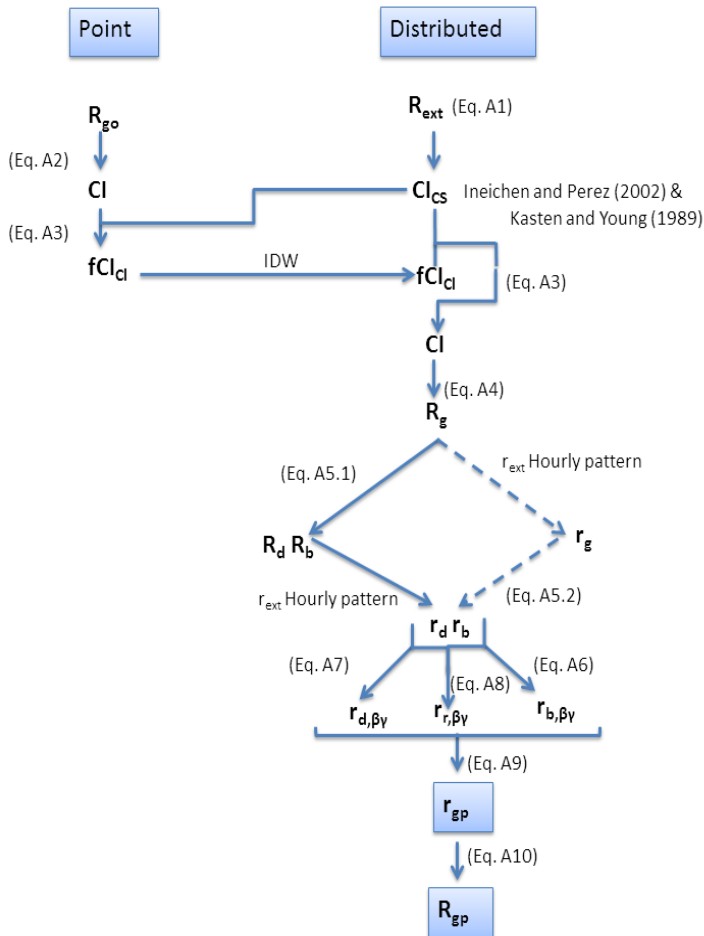


**Figure A1. Flow chart of the solar radiation model**

Hourly beam radiation on a surface of slope β and orientation γ, ($r_{b,\beta\gamma}$ in MJ m$^{-2}$ h$^{-1}$), is calculated according to Eq. A6. in terms of $r_b$, $\theta_z$ and a new corrected zenith angle for the sloping surface, θ (Iqbal, 1983). Then, the model checks the shading effects. Self-shading will occur if the angle between the normal to the surface and the solar vector is greater than 90 degrees. Finally, shading by nearby terrain takes place when the illumination angle is greater than the horizon angle in the same direction. The model previously obtains the horizons following the algorithms of Dozier et al. (1981) and Dozier and Frew (1990), by comparing the slopes between cells in the eight directions.

$$r_{b,\beta\gamma} = r_b \left( \cos\theta \Big/ \cos\theta_z \right)$$

(A6)

Hourly diffuse radiation on a surface of slope β and orientation γ ($r_{d,\beta\gamma}$ in MJ m$^{-2}$ h$^{-1}$), is calculated according to Eq. A7 in terms
of $r_d$ and SVF, the sky view factor, that modifies the incoming radiation incident on a flat surface to consider possibly
obstruction effects on a sloping surface (Dubayah, 1992). Dozier and Frew (1990) obtained an analytical expression for the
estimation of the SVF in terms of the different horizons in each direction considered assuming an isotropic sky.
$$r_{d,\beta\gamma} = r_d \cdot SVF \qquad (A7)$$

Finally, hourly reflected radiation on a surface of slope β and orientation γ ($r_{r,\beta\gamma}$ in MJ m$^{-2}$ h$^{-1}$) and albedo ρ is calculated
according to Dozier and Frew (1990) as expressed in Eq. A8.
$$r_{r,\beta\gamma} = \rho \cdot \left[ \frac{1+\cos\beta}{2} - SVF \right] \cdot (r_d + r_b) \qquad (A8)$$

Hourly global distributed radiation ($r_{gp}$ in MJ m$^{-2}$ h$^{-1}$) is obtained by addition of the three hourly components at each cell
according to Eq. A9.
$$r_{gp} = r_{b,\beta\gamma} + r_{d,\beta\gamma} + r_{r,\beta\gamma} \qquad (A9)$$

Finally, daily global distributed radiation ($R_{gp}$ in MJ m$^{-2}$ day$^{-1}$) is obtained as the summation of hourly global distributed
radiation values (Eq. A10).
$$R_{gp} = \sum_{24h} r_{gp} \qquad (A10)$$


**Appendix B: Comparison with Solar Analyst estimates**
Solar Analyst (SA) is one of the most used GIS-based solar radiation models. It calculates the insolation across a landscape or
for specific locations, based on the methods developed by Fu and Rich (2000a, 2000b, 2002). The total amount of radiation is
given as global radiation and depends on the latitude of the site, topography, shadow cast and atmospheric attenuation. Global
radiation is computed in SA as the sum of direct and diffuse radiation. The equations and modeling scheme can be found in
detail in Fu and Rich (2000b).
Daily global radiation time series were generated in the study site with the Points Solar Radiation tool of SA at each weather
station. Then a cross validation on a leave-one-out process was applied. The main inputs to the model, the diffuse fraction, k,
and the atmospheric transmittivity, τ, were estimated in the study site from observed global radiation data following Batlles et
al. (2008).
SA underestimated observed daily values with a mean α value of 0.78 in the global datasets (Table B.1). In general, worse
approximations to observed data than those shown in Figure 4 were obtained with mean $r^2$ values of 0.66 and RMSE values
ranging from 3.59 (station 853) to 5.11 (station 859) with the global datasets (Table B.1). In terms of the cloudiness level, a
general underestimation by SA was always seen on cloudy (CI≤0.3) and clear-sky days (CI>0.6) with slopes of the fits

significantly lower than 1 (mean α values of 0.42 and 0.74 respectively). In contrast, a slight overestimation with a mean α value of 1.03 was found on partly cloudy days (0.3<CI≤0.6). As for RMSE values, the lowest mean values were always found for cloudy days (1.59 MJ m$^{-2}$ day$^{-1}$), also lower than those obtained in Figure 4 (3.70 MJ m$^{-2}$ day$^{-1}$). However, despite the lower RMSE values the deviation from the 1:1 linear fit in cloudy days with SA estimates was significant (mean α value of 0.42 and r$^2$ value of 0.39 in Table B.1). The highest RMSE values with SA estimates were always found on partly cloudy days with a mean value of 5.35 MJ m$^{-2}$ day$^{-1}$ followed by clear-sky days with a mean RMSE value of 3.37 MJ m$^{-2}$ day$^{-1}$, both considerably higher than those obtained in subsection 3.4 (3.07 and 2.07 MJ m$^{-2}$ day$^{-1}$ respectively).

Table B.1. Model performance with Solar Analyst: slope (α) and r$^2$ of the linear fit between daily predicted *vs.* observed R$_g$ and RMSE (MJ m$^{-2}$ day$^{-1}$)

| Station | | Global data | | | CI≤0.3 | | | 0.3< CI≤0.6 | | | CI>0.6 | | |
|---|---|---|---|---|---|---|---|---|---|---|---|---|---|
| z(m) | Code | α | r$^2$ | RMSE | α | r$^2$ | RMSE | α | r$^2$ | RMSE | α | r$^2$ | RMSE |
| 781 | 602 | 0.77 | 0.69 | 4.05 | 0.27 | 0.43 | 1 | 0.96 | 0.33 | 5.44 | 0.74 | 0.8 | 3 |
| 942 | 608 | 0.73 | 0.74 | 3.78 | 0.23 | 0.46 | 1.06 | 0.86 | 0.54 | 4.56 | 0.7 | 0.81 | 2.97 |
| 950 | 601 | 0.73 | 0.73 | 3.74 | 0.3 | 0.38 | 1.12 | 0.9 | 0.53 | 4.47 | 0.7 | 0.79 | 3.2 |
| 975 | 853 | 0.74 | 0.76 | 3.59 | 0.23 | 0.27 | 1.02 | 0.92 | 0.6 | 4.27 | 0.71 | 0.82 | 2.86 |
| 1212 | 604 | 0.78 | 0.7 | 4.11 | 0.29 | 0.37 | 1.36 | 0.98 | 0.5 | 4.89 | 0.74 | 0.79 | 3.23 |
| 1332 | 803 | 0.75 | 0.71 | 4.07 | 0.38 | 0.5 | 1.27 | 0.99 | 0.46 | 4.89 | 0.72 | 0.8 | 3.2 |
| 1530 | 854 | 0.79 | 0.7 | 4.22 | 0.39 | 0.59 | 1.34 | 1 | 0.49 | 5.12 | 0.74 | 0.8 | 3.11 |
| 1732 | 857 | 0.85 | 0.61 | 4.84 | 0.43 | 0.39 | 1.69 | 1.1 | 0.23 | 6.23 | 0.8 | 0.74 | 3.38 |
| 1735 | 859 | 0.83 | 0.55 | 5.11 | 0.55 | 0.45 | 1.49 | 1.2 | 0.13 | 6.44 | 0.78 | 0.77 | 3.34 |
| 2141 | 804 | 0.67 | 0.68 | 4.02 | 0.24 | 0.17 | 0.95 | 0.97 | 0.5 | 4.72 | 0.64 | 0.79 | 3.21 |
| 2155 | 855 | 0.78 | 0.6 | 4.96 | 0.55 | 0.45 | 1.93 | 1.14 | 0.48 | 5.37 | 0.73 | 0.77 | 3.56 |
| 2300 | 858 | 0.65 | 0.62 | 4.69 | 0.54 | 0.39 | 2 | 0.87 | 0.41 | 5.55 | 0.61 | 0.69 | 4.07 |
| 2325 | 1002 | 0.81 | 0.54 | 4.98 | 0.61 | 0.51 | 2.15 | 1.17 | 0.14 | 6.23 | 0.76 | 0.7 | 3.64 |
| 2510 | 802 | 0.79 | 0.65 | 4.67 | 0.57 | 0.49 | 2.17 | 1.09 | 0.41 | 5.64 | 0.75 | 0.78 | 3.45 |
| 2867 | 1001 | 0.88 | 0.63 | 4.8 | 0.5 | 0.34 | 2.47 | 1.16 | 0.37 | 5.69 | 0.84 | 0.75 | 3.75 |
| 3097 | 860 | 0.85 | 0.6 | 4.95 | 0.58 | 0.08 | 2.45 | 1.22 | 0.24 | 6.1 | 0.81 | 0.74 | 3.87 |
| Mean | | 0.78 | 0.66 | 4.41 | 0.42 | 0.39 | 1.59 | 1.03 | 0.40 | 5.35 | 0.74 | 0.77 | 3.37 |

**Appendix C: Nomenclature**

**Symbols**

CI: daily clearness index

$CI_{CS}$: daily clearness index in a cloudless atmosphere
$CI_h$: hourly clearness index
$E_o$: eccentricity factor
$fCI_{cl}$: cloudiness effects factor
$I_{SC}$: solar constant
k: diffuse to global irradiance ratio
N CI<0.3: rate of days for cloudy conditions
N 0.3<CI<0.6: rate of days for partially cloudy conditions
N CI>0.6: rate of days for clear-sky conditions
$N_o$: number of initially available daily records in the study period
N: number of available daily records after the quality check
$Q_1$: Quartile 1
$Q_3$: Quartile 3
$R_b$: daily beam radiation
$R_{bp}$: daily beam radiation predicted by the model
$R_d$: daily diffuse radiation
$R_{dp}$: daily diffuse radiation predicted by the model
$R_{ext}$: daily extraterrestrial radiation
$R_g$: global radiation
$R_{gcs}$: global radiation under clear-sky conditions
$R_{go\_max}$: maximum daily global radiation observed value
$R_{go\_mean}$: mean daily global radiation observed value
$R_{go\_min}$: minimum daily global radiation observed value
$R_{gp}$: daily global radiation predicted by the model
$r_b$: hourly beam radiation on horizontal surfaces
$r_{b,\beta\gamma}$: hourly beam radiation on a surface of slope β and orientation γ
$r_d$: hourly diffuse radiation on horizontal surfaces
$r_{d,\beta\gamma}$: hourly diffuse radiation on a surface of slope β and orientation γ
$r_{ext}$: hourly extraterrestrial radiation
$r_{r,\beta\gamma}$: hourly reflected radiation on a surface of slope β and orientation γ
$r_g$: hourly global radiation on horizontal surfaces
$r_{gp}$: hourly global radiation predicted by the model
$R^2$: coefficient of determination
$T_L$: Linke turbidity factor
z: elevation

**Abbreviations**
DEM: Digital Elevation Model
IDW: Inverse Distance Weighted
RMSE: Root Mean Square Error
SA: Solar Analyst
SN: Sierra Nevada mountain range
SVF: Sky view factor
UPH: Unsolved Problems in Hydrology

**Greek symbols**
$\alpha$: slope of the fit between $R_{gp}$ and $R_{go}$
$\beta$: slope
$\gamma$: orientation
$\mu_k$: mean of the diffuse to global irradiance ratio
$\rho$: albedo
$\sigma_k$: standard deviation of the diffuse to global irradiance ratio
$\theta$: corrected zenith angle for the sloping surface
$\theta_z$: zenith angle