# Peer review of "Two decades of distributed global radiation time series across a"

_Earth System Science Data, 2020_

## Referee Comment (RC1) · Anonymous Referee #1 · 11 Dec 2020

General comments: The manuscript presents the raster datasets of 19 years of monthly and annual global solar radiation covering the mountainous terrain in Sierra Nevada, Spain with a spatial resolution of 30 meters using a solar radiation model developed in a previous study. While the effort is generally welcome, I have several concerns. 1. The datasets, when compared with most other datasets published on the journal, cover only a very small geographical area, which may significantly limit its use and impact. 19 years of monthly and annual datasets are probably still too short for assessing the trends and shifts in the solar radiation regime. While daily radiation data could be very useful for snow-dominated hydrological modeling in the area, the datasets, unfortunately, don't contain daily scale data. 2. The overall structure

(i.e., sections and sub-sections) is a little bit confusing and need to improve or reorganize. As an example, the same section title, Data Availability, appeared twice in the manuscript. 3. The manuscript spent several paragraphs (for example, section 3.2 and 4.2) on filling data gaps in weather station records and analyzing filled weather station data. Why is this important and necessary? In my opinion, the unfilled weather station data is good enough to generate and validate the datasets.

Specific comments: L13, what does "filled" mean? L13-17, a very long and confusing sentence. Please separate it into several sentences. L18, what does "dispersion" mean here? L19-21, a very confusing sentence. Please rewrite the sentence. L30, I just don't understand ". . . constitute the major when not the only water source for many rivers in the summer" L45-47, Awkward sentence, against interpolation doesn't mean against modeling solar radiation. L57-61, this literature review on GIS-based solar radiation modeling is outdated. L72, what do you mean by "distributed maps"? And 30-m is not really high resolution nowadays. Figure 1. Please indicate that numbers at the stations are their IDs. L108, simply use "Data" L110, what is the source of the DEM? L113-114, this is not a sentence. L113, change "the longest available point information of in situ daily global radiation . . . measured . . ." to "the longest available in situ daily global radiation . . . is measured . . ." L124, "the recorded data" L124-125, what do you mean by "standard limit checking" and "singularities"? L130, prior to? L131, two screenings? L136, what do you mean by "the expression of"? L141, it is not clear how the last screening was performed. L159, what model? Is this the same model used to create the dataset for every cell in the study area? L169, how does the location of weather station affect the modeling result? How the weather station data is used spatially, i.e., which cells use which stations? L170, how do those DEMs affect the modeling result? L170-172, what the purpose of the sentence? L180, N is the number . . . L180-181, this is an interesting interpretation on RMSE. L192-197, Need to provide some details on the spatial and temporal characteristics of the Landsat images used to calculate albedo. L226-228, this is interesting. Are those claims supported by the validation? Please provide the evidence. L237, at each of the? L245, what do you by
"a curved evolution"? L251, Monthly Rg maps L252, what are "the rest of the statistics"? There is no caption on this in Fig. 7. L261, what do you mean by "the monthly distribution of Rg in ..."? L273, very confusing "Monthly distribution of filled daily"! at each of? L286, what are those gray zones? Please explain in the caption. L307, see comment on L273. L316, The second sentence in the caption is very confusing. L325-327, don't understand how the datasets can be used "in other mountainous areas with Mediterranean-type climate conditions and limited radiation station-based observations". L328, How reliable is it to use 19 years of data asses the trends and shifts in the solar radiation regime? L330-332, but those hydrological modeling typically needs daily solar radiation data which are not provided in the datasets. L576, "spatially distributed" → spatially interpolated? L580, Is it possible to directly interpolate CI from the weather stations?

---

## Referee Comment (RC2) · Anonymous Referee #2 · 16 Dec 2020

General comments: The manuscript describes a high spatial resolution global radiation dataset over the Sierra Nevada region in Spain, based on a solar radiation model. Such high-resolution datasets are rare; this is the novelty of the data. My concerns are:

- The applicability of a monthly and annual resolution, though because of missing data in the station data series it is understandable.

- There are many solar radiation models out there. It is not clearly stated why this model is chosen, whether there are better, up-to-date models. I would suggest at least a comparison to other models' skill.

- Why is the daily missing data need to be generated? Since the global radiation has

high variability in mountainous regions, especially in low valleys with fog occurrence, incorporating data based on another station can distort calculations.

- An English language revision is required.

Other specific comments/questions:

- General remark: please refrain from using sentences that are 4-5 lines long, break them up into separate ones.

- L12-16: Too long for one sentence.

- L18, L259,L269: dispersion => use instead scatter or spread, to not cause confusion

- L20: "at the wet season," => in the wet season

- L29-30: Rephrase the second part of the sentence, it is not understandable.

- L30-34: too long sentence

- L73: actor => members

- L80-82 (and L330): Monthly solar radiation data is only suitable for eyeballing surface energy budget components, and most definitely won't help with runoff in a mountainous area.

- L93: end of sentence dot is missing

- L94-94: please, rephrase the sentence with a different word structure.

- L95-97: I don't understand the sentence.

- Figure 1.: Please, note in the caption that numbers on the figure at the station IDs.

- L109-110: Which specific DEM model is used?

- L113-114: change the sentence, from:" the longest available point information of in situ daily global radiation (Rgo) measured in 16 weather stations over the area", to "the

longest in-situ daily global radiation (Rgo) of 16 weather stations over the area"

- L114-115: There are only 4 low altitude stations in the first 5 years of the data set. How reliable the global radiation estimation is in this case?

- L141: What is the exact "expectancy envelope" in this case?

- L162: How is the cell defined? The 30x30 m grid point, or is it a larger one?

- L180-181: Please, check the definition of the RMSE.

- L189-190: The original model differentiates the diffuse radiation estimates at CI<0.1, 0.1 <=CI <0.71, and CI>0.71. Is using these intervals affect or change the interpretation of the results?

- L189: "atmospheric states" => an atmospheric state is stable, unstable or neutral, => "based on the cloudiness three types of weather conditions were analysed: . . ."

- L193-194: What periods the satellite measurements cover? It would be informative to give the horizontal resolution of the satellite images.

- L202-203: A high correlation coefficient is expected since the global radiation has a clear intra-annual course. Instead of a simple linear correlation for the whole dataset, the annual course should be removed and then calculate the correlation.

- Figure 3, Figure 4: Beside the station IDs, the altitude of the stations could also be shown, so one doesn't have to scroll back-and-forth to analyse the figure on their own. Or the figures/columns could be ordered by altitude, so it would be more informative as there is no seemingly order in the current figures/columns.

- L225-228: Station 853 has lower RMSE than 802 or 860, though it is situated far away from the other stations, so the "leave one out method" for validation would affect it the most. (comparing 858 and 860, which are both high altitude station, the verification scores are still worse for 860 which is surrounded by 3 other stations) How is the statement in these lines are then supported?

- L231, L233,L292,L299 (I'm sure I left some out): "appreciated" is not the correct word to be used here, perhaps use "shown" instead.

- L230-235: too long sentence

- L231-232: a stable minimum value is attributed to the occasional cloudy days, it is expected to have low variations.

- L245: I get what you mean by curved evolution, but it should be rephrased as it means something different

- L245: Why is there a difference between the curves from January to July, and from August to December? (it only looks linear because of the temporal resolution, but the second semester's global radiation are lower than the first. Is it because of precipitation?)

- L258: "allow to draw the same conclusions as those" Was it assumed to be otherwise?

- L262: Which months constitute the "wet season"?

- Figure 6: The color scale is not fortunate in terms of values. Using the same scale for winter and summer months is not a good idea as low radiation values disappear from the map. Perhaps use two scales, one for the winter semester and one for the summer semester.

- Figure 8: What is the grey area mean? What do the different grayscale colours mean? The timeseries is too long for a good figure. Variations in the data can barely be observed. It would be better to split the figure into two periods.

- Figure 11: Do the grayscale colours correspond to the percentiles? If so, please note it in the caption.

- L319-327: The paragraph refers in general long term solar radiation data, but one should be careful with it, and highlight the ones that are representative to this particular dataset.

- L320-321: To analyse the Sun's activity on a barely 17 year-long dataset is a far reach.

- L326: The region covered by the dataset is roughly 50 km by 100 km, it is definitely not a large scale when it comes to atmospheric processes. It might affect large-scale processes such as extratropical cyclones (change in direction or intensity) but only due to its orographic properties and due to radiation properties.

- L576,L577: distributed => interpolated?

---

## Author Comment (AC1) · 21 Jan 2021

We would like to acknowledge and thank the Reviewers and Editor for their work and useful and interesting comments, which were really helpful to improve the manuscript.

We are attaching 3 documents:

- 1_Response to Reviewers.pdf: a point-by-point answer to both Reviewers, in which reference to modifications in the paper is included when needed as lines/pages/sections in the revised manuscript (also attached).

- 2_Revised_Manuscript.pdf: revised version of the manuscript (new or corrected text

in yellow).

- 3_New_Manuscript.pdf: new version of the manuscript.

Regards,

/The authors

Please also note the supplement to this comment:
https://essd.copernicus.org/preprints/essd-2020-250/essd-2020-250-AC1-supplement.zip
* * *

---

## Author Comment (AC2) · 21 Jan 2021

Please, find in the supplement to the AC1 our reply

Regards,

The authors

—————————————————————

---

## Author Comment (AC3) · 21 Jan 2021

Please, find in the supplement to the AC1 our reply

Regards,

The authors

––––––––––––––––––––––––––––––

---

## Author Response (AR1)

We would like to acknowledge and thank the Editor and the Reviewers for their valuable comments and issues raised in order to strengthen the value of the study. We offer a detailed point-by-point discussion here below where the original comments by the reviewers (indicated with "R") are copied in italic. We also present how we have addressed the Reviewers' comments ("C"). Reference to modifications in the paper is included when needed as lines (LX) in the revised manuscript.

**#Reviewer 1**

General comments: The manuscript presents the raster datasets of 19 years of monthly and annual global solar radiation covering the mountainous terrain in Sierra Nevada, Spain with a spatial resolution of 30 meters using a solar radiation model developed in a previous study. While the effort is generally welcome, I have several concerns.

**C1** 1. The datasets, when compared with most other datasets published on the journal, cover only a very small geographical area, which may significantly limit its use and impact.

**R1** We understand the reviewer's concern regarding the potential use and impact of the datasets. However, we think that the geographical extension of a study site is not the main factor that determines these aspects. In fact, there are several studies already published in this journal providing meteorological data for hydrological studies that cover smaller areas than the 4583.72 km2 area of the present study (e.g., Nord et al., 2017; Bales et al., 2018; Fang et al., 2019). There are also global studies that cover large extensions of the Globe but their spatial and/or temporal resolutions limit the number of potential users working at local scales. Instead, we think that the use and impact of datasets depends on the combination of both the spatial and temporal resolutions of the data provided together with the number of actors with potential interest on the datasets.

In Sierra Nevada (SN) there are numerous members that often require global radiation estimations with very varied purposes (e.g., floods and droughts forecasts, irrigation scheduling activities, snow projections in the sky resort, building services design engineering, etc.). Also, the scientific community working on the estimation of solar radiation in mountainous areas agrees that one of the main drawbacks is the lack of reliable observed data. However, both local and international potential users have in common that the spatial resolution of available solar radiation datasets becomes a major issue given the great spatial heterogeneity of these mountainous areas.

Therefore, and once the data repository has been updated (please see R3), we believe that the biggest impact of the datasets provided in this study relies on two aspects. The first one is the combination of both high spatial (30 m) and temporal resolutions (daily) of the maps provided along 19 years. Thus, its use in the future is guaranteed given the numerous local members interested in these datasets in SN as stated before. The second one is the open access to datasets in a mountainous area such as SN. SN is a relevant site for climate and biodiversity research that attracts different international teams as the most southern point in the EU where there is snow snowfall and recognized by different international environmental protection figures (e.g., climate change observatory for mountain areas in the international network, UNESCO biosphere reservoir). Moreover, dense and properly maintained weather station networks in mountainous areas are rarely available, so researchers involved in high mountain solar radiation estimation constitute the second group of potential users of these data, which can be further used for calibration/validation purposes of different algorithms, comparison to other similar areas, etc.

**C2** 19 years of monthly and annual datasets are probably still too short for assessing the trends and shifts in the solar radiation regime.

**R2** We completely agree with the reviewer that the study period is not enough to precisely assess trends and shifts in the solar radiation dynamics. Therefore, we have modified and deleted those statements all through the revised manuscript (e.g., L22, L394).

**C3** While daily radiation data could be very useful for snow-dominated hydrological modeling in the area, the datasets, unfortunately, don't contain daily scale data.**

**R3** Even though we already had computed daily radiation map series to generate the monthly and annual datasets, we regret not having provided them in the previous version. Nevertheless, as we fully agree that daily radiation data is essential for hydrological studies, and following both reviewer's suggestions we have included the daily datasets in the revised version of the manuscript (L9, L24, L79, L265-286, L367-375, L377) as well as in the data repository, being totally available at: <a href="https://doi.pangaea.de/10.1594/PANGAEA.921012">https://doi.pangaea.de/10.1594/PANGAEA.921012</a>.

Moreover, we would like to point out that hourly radiation data can be provided by the authors as well. This temporal scale is highly required when modelling the snow dynamics, something remarkable in alpine catchments like the study site. We did not add these data to the data repository due to two reasons. The first one is the large storing capacity requirements. The second one is that a specific validation of these hourly datasets like the one applied in the daily estimates at weather stations has not been carried out in this study. The only available validation of hourly datasets in SN was carried out in Aguilar et al. (2010) at station 802 and for a shorter study period (2004-2010). Thus, we added the following statement at the end of section 5 (L368-373): "Hourly datasets were also computed in this study but due to their large storing capacity requirements they have not been included in the data repository specified above. Thus, hourly maps can be provided for certain dates upon request to the authors. However, a validation of these hourly datasets like the one applied in the daily estimates at the weather stations has not been specifically carried out in this study. Therefore, in case hourly maps are requested to the authors, these data should be taken with caution as the only available validation in SN was carried out at one weather station (802 in Fig. 1) and for a shorter period (2004-2010) in Aguilar et al. (2010)."

**C4** 2. The overall structure (i.e., sections and sub-sections) is a little bit confusing and need to improve or reorganize. As an example, the same section title, Data Availability, appeared twice in the manuscript.

**R4** We apologize for the duplication in the section title and following the reviewer's comment we have restructured the manuscript as follows:

- 1 Introduction
- 2 Study site
- 3 Data
- 3.1 Input data
- 3.2 Data quality control
- 3.3 Generation of global radiation data
- 3.4 Cross-validation at weather stations

4 Results

- 4.1 Daily time series of global radiation in Sierra Nevada
- 4.2 Monthly time series of global radiation in Sierra Nevada
- 4.3 Annual times series of global radiation in Sierra Nevada
- 5 Data availability
- 6 Final remarks

**C5** 3. The manuscript spent several paragraphs (for example, section 3.2 and 4.2) on filling data gaps in weather station records and analyzing filled weather station data. Why is this important and necessary? In my opinion, the unfilled weather station data is good enough to generate and validate the datasets.

**R5** First of all we would like to apologize for the confusion created with the use of the "filling" term as we are aware that the explanation of the map generation sequence was not clear enough in the previous version of the manuscript. Secondly, we would like to point out that both, the validation process, and the generation of the datasets in the previous version, were carried out with the observed ("unfilled") available datasets once the quality-check was applied. With filled daily datasets we meant the complete daily radiation data set at weather stations with no gaps often required for simulation purposes (Muneer and Gul, 2000). These missing values at pixels where a weather station is located are modelled as any other pixel within the DEM. Thus, this continuous daily dataset at weather stations contains both the observed data and the modeled values when there are gaps at each weather station. The reason to include the analysis of filled daily weather station data (former section 4.2) was to complement the validation analysis data through the agreement with the statistics of observed data (Table 1). The model takes as input daily Rg data just the observed available values and not this "filled data", so results regarding the validation and the generation of datasets are not affected. However, in view of both reviewers' comments we have tried to improve the explanation of the solar radiation generation process with no reference to fill/unfilled data in the revised version (e.g., Sections 3.3 and 4.1). Besides, with the new re-organization of the manuscript suggested in C4 we hope to have gained some clarification in the matter. Finally, with the inclusion of the daily datasets in the new version we have added as a supplement to the manuscript a .txt file indicating gaps in observed Rg values per day and weather station as indicated in section 5 (L366-367). Thus, potential users of these datasets can check whether for a certain day and weather station the daily Rg value in the pixel where it is located is either measured (code 1) or modelled (0).

**Specific comments:**

**C6** L13, what does "filled" mean?**

**R6** Please see response R5 above where we explained what we meant with the filled data expression in the original version of the manuscript.

**C7** L13-17, a very long and confusing sentence. Please separate it into several sentences.**

**R7** We acknowledge the suggestion and thus, we have modified the sentence as follows (L14-16): "Daily  $R_g$  at weather stations revealed greater variations in the maximum daily  $R_g$ , but no clear trends with altitude in any of the statistics. However, at the monthly and annual scales, there is an increase in the high extreme statistics with the altitude of the weather station, especially above 1500 m a.s.l."

**C8 L18, what does "dispersion" mean here?**

**R8** Following reviewer's 2 suggestion (please, see C52) we replaced "dispersion" with "scatter" (L18, L304 and L329).

**C9** L19-21, a very confusing sentence. Please rewrite the sentence.**

**R9** We have rewritten the sentence as follows (L19-21): "The monthly  $R_g$  distribution was highly variable along the study period (2000-2018). Such variability, especially in the wet season (October-May), determined the inter annual differences of up to 800 MJ m-2 year-1 in the incoming global radiation in SN."

**C10** L30, I just don't understand ". . . constitute the major when not the only water source for many rivers in the summer"**

**R10** We have rephrased the sentence as follows (L32-33): "They play a key role as water providers during the warm and dry season when they often constitute the only water source for many rivers."

**C11** L45-47, Awkward sentence, against interpolation doesn't mean against modeling solar radiation.

**R11** For a better understanding we have modified the sentence as (L48-49): "All of them insist on the need to consider topographic effects and advise of the errors that simple interpolation/extrapolation techniques can create."

**C12** L57-61, this literature review on GIS-based solar radiation modeling is outdated. **R12** Following the reviewer's comment we updated the review (L54, L66) with recently published studies by Zhang et al. (2019 and 2020).

**C13** L72, what do you mean by "distributed maps"? And 30-m is not really high resolution nowadays.**

**R13** We apologize for the redundancy and we just left "maps" in the sentence (L79) Among the two main methodologies for solar radiation modeling stated in the manuscript (L60-64), satellite-derived solar radiation models provide a wide spatial and temporal coverage, but coarse spatial resolutions when dealing with pixels with a strong topographic gradient. In fact, up to our knowledge the finest spatial scale of studies that derive global radiation estimations from satellite observations ranges from 0.05-0.5° (e.g., Tang et al., 2019; Hao et al., 2020). As for GIS-based solar radiation models, finer spatial scales can be achieved and thus they are more suitable than the former to capture local variations in mountainous areas, but they are very computationally demanding (e.g., Zhang et al., 2020). Thus, applications ranging from 20 m for small study sites (e.g., Tovar-Pescador et al., 2006; Batllés et al., 2008; Ruiz-Arias et al., 2009), to 1 km for large territorial extensions (e.g., Roupioz et al., 2016; Zhang et al., 2020) are found in the literature.

We would like to point out that even though a higher resolution DEM (10-m) was available in the study site, albedo was estimated through Landsat images with a 30 m spatial scale. Larger spatial resolution data sources for the albedo estimation (e.g., Sentinel 2 data) do not reach the first decade of the study period. Thus, we selected 30 m to match the spatial scales of both data sources considering the computation requirements determined by the geographical extension of the study site, the length of the study period, and the temporal scale of the generated datasets. Thus, it is the combination of these aspects that make us think that 30 m is a high spatial resolution within the scope of these types of studies.

**C14** Figure 1. Please indicate that numbers at the stations are their IDs.**

**R14** We changed the caption into (L113-114): Figure 1. Location of the study site in southern Spain (left). Digital Elevation Model (DEM) and weather stations in Sierra Nevada (SN) (right). The numbers correspond to the station codes.

**C15** L108, simply use "Data" **R15** Done (L115).

**C16** L110, what is the source of the DEM?**

**R16** The digital elevation model was provided by the Andalusian regional administration. It was generated by digital stereo correlation of aerial photographs of the Spanish National Plan of Orthophotography included in the National Aerial Orthophotography Plan (PNOA). We added such information in the revised version (L117-119).

**C17** L113-114, this is not a sentence & L113, change "the longest available point information of in situ daily global radiation . . . measured . . ." to "the longest available in situ daily global radiation . . . is measured . . ."

**R17** Following both reviewers' comment We replaced that sentence with (L121): "Meteorological input data are the longest available in-situ daily global radiation ( $R_{go}$ ) of 16 weather stations over the area."

**C18** L124, "the recorded data"

**R18** We acknowledge this correction (L132).

**C19 L124-125, what do you mean by "standard limit checking" and "singularities"?**

**R19** With standard limit checking we meant that the observed global radiation at weather stations must range between the clear daily global radiation ( $R_{gcs}$ ) and 3% of the daily extraterrestrial radiation ( $R_{ext}$ ). Thus, the two first screening described in section 3.2 (L140-145). As for singularities we meant some particularities often found at weather stations in high altitudes that cause operational errors in the measurements (e.g., shadows, impacts of snow, mechanical failures due to extreme meteorological conditions, etc.).

However, after the new reorganization detailed in R4 this statement was simplified and integrated in section 3.2.(L133-165).

*C20 L130, prior to?* **R20** We acknowledge this correction (L137).

**C21** L131, two screenings?

**R21** Accordingly, we replaced logical tests with screenings (L138).

**C22 L136, what do you mean by "the expression of"?**

**R22** We mean that  $R_{gcs}$  values were calculated with the model or equation developed by Ineichen and Perez (2002) (Equation 11). For a better understanding we rephrased the sentence as (L143-145): " $R_{gcs}$  values were calculated with the model developed by Ineichen and Perez (2002) and the parameterization of Kasten and Young (1989) for the air mass. More detail regarding the equation as well as its parameters can be found in Aguilar et al. (2010)."

**C23** L141, it is not clear how the last screening was performed.**

**R23** The last screening was performed following the specifications of Younes et al. (2005) for the creation of the expectancy envelopes in the CI-k chart. To clarify the implementation of this screening the following paragraphs were added (L150-161): "The CI data range is divided into bands of equal width, within which the mean and standard deviation of the k values,  $\mu_k$  and  $\sigma_k$ , are calculated. The top and bottom boundary shapes are identified by fitting two polynomials through the points  $\mu_k \pm b\sigma_k$  limited between 0 and 1 to respect the physical range of the CI. In this study b values between 2 and 3 were applied in order to limit both, the rejection of good data and the acceptance of erroneous data to small percentages.

The CI was calculated with the observed data at each weather station. However, no measurements of daily diffuse radiation, Rd, were available. Thus, the model proposed by Aguilar et al. (2010) was applied to generate daily diffuse radiation ( $R_{dp}$ ) at each weather station without considering the observed global data at such station. Obviously, this assumption depends on the validity of the model as well as on the quality of  $R_{go}$  datasets at the remaining weather stations. However, under the common lack of diffuse solar radiation measurements like the present one, modeling them can be an alternative (e.g., Yang et al., 2020) to reject erroneous  $R_g$  observations. This approach was proposed once the model had already been validated in a previous study (Aguilar et al., 2010) but keeping in mind the intrinsic limitations and assumptions previously stated."

**C24** L159, what model? Is this the same model used to create the dataset for every cell in the study area?

**R24** Yes, it is. Please see R5 for the complete answer to this question. In the new version this part of the manuscript has been removed.

*C25* L169, how does the location of weather station affect the modeling result? How the weather station data is used spatially, i.e., which cells use which stations?

**R25** As in any interpolation algorithm, the more representative of the global heterogeneity of the target area the initial set of stations is, the more accurate the modelled result will be. Please, see in R64 a simulation with limited input data where the effect of the spatial configuration of the weather station network can be assessed.

Figure A1 summarizes the solar radiation simulation scheme further developed in Aguilar et al. (2010). Observed  $R_g$  data are used to compute the clearness index (CI) at each weather station (named Point in Fig. A1). This value is interpolated through IDW. The limitations of this step of the modeling process are fully discussed in Aguilar et al. (2010) as well as latter in R45.

**C26 L170, how do those DEMs affect the modeling result?**

**R26** The DEM resolution plays a very important role in obtaining accurate solar radiation estimates. We have not run the model with those DEMs as their spatial resolution is too coarse to properly capture the spatial heterogeneity of SN. Nevertheless, different works have assessed how increasing the cell size leads to larger mean error in GIS-based solar radiation estimates relative to in-situ measurements (e.g., Ruiz-Arias et al., 2009; Zhang et al., 2020).

**C27** L170-172, what the purpose of the sentence?**

**R27** Following this comment, we deleted the sentence as we understand that potential readers of the manuscript are familiar with open access GIS data.

**C28** L180, N is the number . . .

**R28** We appreciate this correction (L205).

**C29** L180-181, this is an interesting interpretation on RMSE.**

**R29** According to both reviewers' comments we changed the definition of the RMSE as (L205-206): "It measures the difference between values predicted by the model and those which were actually observed."

*C30* L192-197, Need to provide some details on the spatial and temporal characteristics of the Landsat images used to calculate albedo.

**R30** Following both reviewers' comments we have added more detail concerning Landsat images within the manuscript (L192-198). Moreover, a new figure (Figure 3) has been incorporated with the specification of dates and sensors of the images analyzed in this study.

**C31** L226-228, this is interesting. Are those claims supported by the validation? Please provide the evidence.**

**R31** The validation results (Figure 4) support the conclusion that there is no clear pattern of the goodness of the model estimates with the height of a certain weather station. In fact, it is the interaction of different aspects that determine the reliability of estimated Rg values. Please, see R73 for a further elaboration of this response.

**C32 L237, at each of the?**

**R32** We acknowledge this correction. Changes have been applied in the captions of Figures 4, 5 7 and 11 in the revised version.

**C33 L245, what do you by "a curved evolution"?**

**R33** We replaced "curved evolution" by "slightly convex evolution" (L291).

**C34 L251, Monthly Rg maps**

**R34** We acknowledge this correction. Changes have been applied in several sentences of the revised version (L297, L342).

**C35 L252, what are "the rest of the statistics"? There is no caption on this in Fig. 7.**

**R35** We had to re-number several figures to include some new figures after the revision process so former Fig. 7 is Fig. 9 in the new version. We apologize for the error and thus, we have completed the caption as follows (L318-319): "Figure 9. Statistical distribution of the monthly  $R_g$  values throughout the study area. Whisker boxes represent the 10th, 25th, 50th, 75th and 90th percentiles of each monthly map per year."

**C36 L261, what do you mean by "the monthly distribution of Rg in ..."?**

**R36** We mean for each monthly map generated, the statistical distribution of  $R_g$  in terms of the following percentiles:  $10^{th}$ ,  $25^{th}$ ,  $50^{th}$ ,  $75^{th}$  and  $90^{th}$ . We hope to have clarified it with the new Figure 9 caption.

**C37** L273, very confusing "Monthly distribution of filled daily"! at each of?**

**R37** Here again we apologize for the mess explained in R5. Accordingly, we renamed it as (L308): "Statistical distribution of monthly  $R_g$  (MJ m-2 month-1) time series".

**C38** L286, what are those gray zones? Please explain in the caption.**

**R38** The grey zones in both Figures 10 and 13 represent the following percentiles: 5th, 10th, 25th, 50th, 75th, 90th and 95th. We have completed the captions of both figures for a better clarification. Also, following reviewer's 2 suggestion (C82) we split figure 10 into two periods for a better visualization of the variations.

**C39* L307, see comment on L273.**

**R39** As in R37 we renamed it as (L347): "Statistical distribution of annual  $R_g$  (MJ m-2 year-1) time series".

**C40** L316, The second sentence in the caption is very confusing.**

**R40** As in R38, grayscale zones represent the following percentiles:  $5^{th}$ ,  $10^{th}$ ,  $25^{th}$ ,  $50^{th}$ ,  $75^{th}$ ,  $90^{th}$  and  $95^{th}$ . Dashed lines represent the mean value of each percentile in the study period. Thus, the mean  $50^{th}$  percentile value of annual Rg in SN is close to  $6500 \text{ MJ m}^{-2}$  year-1. For a better understanding, we have added a legend in the Figure and rewritten the caption as follows (L364-365): "Figure 13. Evolution of the statistical distribution of annual Rg (MJ m-2 year-1) in the study period (2001-2018) throughout the study area. Dashed lines represent the mean values of the percentiles analyzed."

**C41** L325-327, don't understand how the datasets can be used "in other mountainous areas with Mediterranean-type climate conditions and limited radiation station-based observations".**

**R41** We meant that spatial and temporal variability from these data sets can shed light on the most relevant factors affecting the heterogeneity of solar radiation in abrupt topography in these areas. Also, they can help to estimate the order of magnitude of the variation range, their relationship with slope, orientation, altitude, etcetera. Being limited as this might be, the analysis would provide some estimation of uncertainty when estimating average values from scarce weather monitoring networks, or short time series. We agree that as it is written it is much to be said and have modified this accordingly as (L401-404): "These results can also assess the order of magnitude of different sources of spatial variability (altitude/slope/aspect gradients) as well as the seasonal range of variation at different time scales and their annual variability. This estimation may provide a first estimate of the order of magnitude of uncertainty

of average calculations or spatial interpolation from a scarce number of weather stations in Mediterranean and semiarid mountain areas."

**C42** L328, How reliable is it to use 19 years of data asses the trends and shifts in the solar radiation regime?

R42 Please, see reply in R2.

**C43** L330-332, but those hydrological modeling typically needs daily solar radiation data which are not provided in the datasets.

**R43** We hope to have solved this with the inclusion of daily data. Please, see reply in R3.

**C44** L576, "spatially distributed"  $\rightarrow$  spatially interpolated? **R44** We agree with this suggestion, and we did the replacement in L638 and L639.

**C45** L580, Is it possible to directly interpolate CI from the weather stations **R45** The model was originally developed to be run with minimum input data requirements: DEM, albedo and Rgo, so cloud accounting had to be estimated by directly interpolating CI. Further discussion on the matter can be found in Aguilar et al. (2010).

We are aware that the most challenging issue in solar radiation modeling in data sparse regions is cloud accounting, due to the rapid spatially and temporally changing weather conditions and the three-dimensional structure of clouds. In fact, as Zhang et al. (2020) recently stated: "a GIS-based solar radiation model that allows for the treatment of high spatial and temporal variability in sun-earth position, terrain, and atmospheric effects has not yet been developed for monitoring daily solar radiation.". Therefore, further research is continuously being carried out by the authors to better quantify radiative effects of clouds from easily available data sources. Here, with the rapidly rising array of satellite products some atmospheric products are an important asset for future research as long as they have the required spatial scale in the study.

**#Reviewer 2**

General comments: The manuscript describes a high spatial resolution global radiation dataset over the Sierra Nevada region in Spain, based on a solar radiation model. Such high-resolution datasets are rare; this is the novelty of the data. My concerns are:

We would like to thank Reviewer #2 very much for this appreciation.

**C46** - The applicability of a monthly and annual resolution, though because of missing data in the station data series it is understandable.

**R46** The availability of data at these two scales was a matter of our thought regarding their potential use by other users. The monthly and annual maps were calculated from daily information maps and hence, they were already computed. Following both reviewers' comments (Please, see answer R3) we have decided to make them also available at: https://doi.pangaea.de/10.1594/PANGAEA.921012

**C47** - There are many solar radiation models out there. It is not clearly stated why this model is chosen, whether there are better, up-to-date models. I would suggest at least a comparison to other models' skill.

**R47** We completely agree that a comparison to at least another available solar radiation model demonstrates the best suitability of a certain model in a study site. Thus, the analysis with Solar Analyst estimates in SN has been included as an appendix in the revised version (Appendix B). Moreover, a discussion of the comparison to previous studies that applied other more-data

demanding GIS-based models in a small sub-area (10x5 km2) within the north-eastern side of SN is included in section 3.4 (L236-253).

With both analyses we hope to have justified the choice of the model (as proposed in section 3.3) for generating global radiation datasets at the spatial scale analyzed in this study site.

**C48** - Why is the daily missing data need to be generated? Since the global radiation has high variability in mountainous regions, especially in low valleys with fog occurrence, incorporating data based on another station can distort calculations.

**R48** The data filling done is not a proper data filling. Here again we apologize, and further explanation of this comment can be found in R5.

**C49 - An English language revision is required.**

**R49** We have tried to solve major concerns about clarity in language, accepting all reviewers' corrections, suggestions and addressing the comments below. We have specifically checked English guidelines and house standards from the journal. Moreover, the manuscript has been checked by a native English speaker trying to improve the language. In addition, ESSD applies an English language copy-editing before sending the manuscript galley proof. In any case, if Reviewer #2 still does not find clear enough the language, we will contact a different native speaker for a second check.

**Other specific comments/questions:**

**C50** - General remark: please refrain from using sentences that are 4-5 lines long, break them up into separate ones.

**R50** Accordingly, we have carried out a deep revision of the manuscript to make it more readable. Apart from the corrections indicated by both reviewers (e.g., C51, C54, C55, C75, etc.), we have rephrased some other sentences along the manuscript.

**C51 - L12-16: Too long for one sentence.**

**R51** We have modified the sentence in the revised version (L12-16) as follows: "The applicability of the modeling scheme was validated against daily global radiation records at the weather stations. Mean RMSE values of 2.63 MJ m-2 day-1 and best estimations on clear-sky days were obtained. Daily Rg at weather stations revealed greater variations in the maximum daily Rg, but no clear trends with altitude in any of the statistics. However, at the monthly and annual scales there is an increase in the high extreme statistics with the altitude of the weather station, especially above 1500 m a.s.l."

**C52** - L18, L259,L269: dispersion => use instead scatter or spread, to not cause confusion. **R52** We acknowledge the suggestion and we replaced "dispersion" with "scatter" (L18, L304 and L329).

C53 - L20: "at the wet season," => in the wet season.R53 Change applied in L20.

**C54** - L29-30: Rephrase the second part of the sentence, it is not understandable.**

**R54** We rephrased the sentence as follows (L32-33): "They play a key role as water providers during the warm and dry season when they often constitute the only water source for many rivers".

**C55** - L30-34: too long sentence.**

**R55** We have modified the sentence in the revised version (L33-36) as follows: "Here, water fluxes from the snowpacks show a shift from the predominant partition between snowmelt and sublimation usually found in colder and wetter climates on an annual and seasonal basis

(Herrero and Polo, 2016). This shift is caused by the radiation balance that enhances sublimation during cold and dry periods and intense snowmelt rates during late winter and spring in these areas (McDonell et al., 2013; Liu et al., 2019)".

**C56** - L73: actor => members. **R56** Change applied in L80.

**C57** - L80-82 (and L330): Monthly solar radiation data is only suitable for eyeballing surface energy budget components, and most definitely won't help with runoff in a mountainous area. **R57** We completely agree with the reviewer. The response to this comment can be found in R3.

**C58 - L93: end of sentence dot is missing.**

**R58** We apologize for the error and we made the right punctuation in L101.

**C59** - L94-94: please, rephrase the sentence with a different word structure.**

**R59** We rephrased the sentence as follows (L101-102): "The snow presence becomes relevant from November above 2000 m a.s.l. and extends up to spring with conditions that make it possible the activity of a major ski resort in the area."

**C60* - *L*95-97: *I* don't understand the sentence.**

**R60** We reformulated if as follows (L102-103): "However, in some winters, mild episodes can be found in January and February that melt most of the snow much earlier than the mean end of the snow season in the area (Herrero et al., 2009; Herrero and Polo, 2012)."

**C61** - Figure 1.: Please, note in the caption that numbers on the figure at the station IDs.**

**R61** We changed the caption into (L113-114): Figure 1. Location of the study site in southern Spain (left). Digital Elevation Model (DEM) and weather stations in Sierra Nevada (SN) (right). The numbers correspond to the station codes.

**C62 - L109-110: Which specific DEM model is used?**

**R62** The digital elevation model was provided by the Andalusian regional administration. It was generated by digital stereo correlation of aerial photographs of the Spanish National Plan of Orthophotography included in the National Aerial Orthophotography Plan (PNOA). We added such information in the revised version (L117-119).

**C63** - L113-114: change the sentence, from:" the longest available point information of in situ daily global radiation (Rgo) measured in 16 weather stations over the area", to "the longest insitu daily global radiation (Rgo) of 16 weather stations over the area" **R63** Accordingly, we did the replacement in L121.

**C64** - L114-115: There are only 4 low altitude stations in the first 5 years of the data set. How reliable the global radiation estimation is in this case?**

**R64** In order to answer this comment we calculated the cross-correlation analysis for the whole study period when only the four low altitude stations (601, 602, 604 and 608 in Fig. 1) were used as inputs to the model. The results are shown in Figure R1. As it was expected, RMSE values increased at every station with a magnitude that depends on the cloudiness level. It is remarkable that there is a significant increase of over 2 MJ m-2 day-1 in the RMSE values of cloudy days at several stations compared to those shown in Figure 4 (page 11). The increase in RMSE values in clear-sky days and in the global data was also very variable among the stations but not higher than 0.5 MJ m-2 day-1 in most of them.

The reliability of global radiation estimates in the first five years of the study period is obviously lower than in the rest of the period. However, the errors obtained under this limited input data

---

## Author Response (AR2)

Once again, we would like to acknowledge and thank the Editor and Reviewers for their work and their useful and interesting comments. We strongly believe that the feedback has been really fruitful, and that the manuscript and datasets have greatly improved along the revision process. We offer here below in italic the last reflections by reviewer 2 and present how we have addressed them. Reference to modifications in the paper is included when needed as lines (LX) in the revised manuscript.

*Reflections to Author's responses:*

*C64: I agree that the errors are still within an acceptable margin. However, since the purpose of the journal is to describe datasets, in the light of the results you provided in the response, it is worth mentioning that in the first 5 years higher elevation stations are subjected to a slightly greater overestimation of solar radiation, especially during cloudy conditions. And add few indicator numbers such as the Δα, and ΔRMSE range.*

*C71: Comparability to previous studies is important, I accept that you'd wish to publish the correlations without deseasonalisation. For the sake of accuracy, you should also mention the results you got with deseasonalisation. Something along the lines of, comparison of deseasonalised data show a higher accuracy of the model ... and add the Δα, and ΔRMSE range. Maybe even adding those results as a supplementary figure is also worthwhile.*

We agree that both results are interesting and therefore integrated them within the manuscript in section 3.4 and Appendix B. In this way the following statements were included:

L217-223: " The cross-validation analysis was also carried out with deseasonalized daily data to remove the expected intra-annual course of global radiation data. The deseasonalization of the daily series was carried out applying a stable seasonal filter (Brockwell and Davis, 2002) as already done in a previous study with other hydrometeorological datasets (Aguilar et al., 2017). Besides, as the reliability of solar radiation estimates is conditioned by the availability of recorded data, the cross-validation analysis for the whole study period was also computed with limited data. Thus, global radiation estimated were generated with only the four stations (601, 602, 604 and 608 in Fig. 1) with the longest records (Figure 2) as inputs to the model. Results are shown in Appendix B."

L246-254: "With the deseasonalized time series (Fig. B1), differences were reduced among the different cloudiness levels. The most remarkable change was a significant improvement in the estimates of cloudy days in every station when the range of RMSE values shifted from 2.54-7.52 (in red in Fig. 4) to 1.72-5.16 MJ m$^{-2}$ day$^{-1}$ (in red in Fig. B1). Also, the range of the slopes significantly narrowed from 1.18-1.74 (red α values in Fig. 4) to 0.92-1.09 (red α values in Fig. B1). Thus, the comparison with deseasonalized data showed a higher accuracy of the model than the one obtained with the original datasets (Fig. 4).

The comparison with limited input datasets shown in Figure B2 confirmed the lower reliability of global radiation estimates in the first five years when datasets recorded at only four stations (601, 602, 604 and 608 in Fig. 1) were available in SN. Here, higher elevation stations are subjected to a slightly greater overestimation of solar radiation (1.34-2.04 in red in Fig. B2), especially during cloudy conditions when the RMSE values increased to 3.62-8.45 MJ m$^{-2}$ day$^{-1}$ (in red in Fig. B2)."